# Seasonal trends of Ice Nucleating Particles at Ny-Ålesund: a study of condensation-freezing by the Dynamic Filter Processing Chamber

Matteo Rinaldi<sup>1</sup>, Alessia Nicosia<sup>1</sup>, Marco Paglione<sup>1</sup>, Karam Mansour<sup>1</sup>, Stefano Decesari<sup>1</sup>, Mauro Mazzola<sup>2</sup>, Gianni Santachiara<sup>1</sup>, Franco Belosi<sup>1</sup>

<sup>1</sup>Institute of Atmospheric Sciences and Climate, National Research Council of Italy, Bologna, 40129, Italy <sup>2</sup>Institute of Polar Sciences, National Research Council of Italy, Bologna, 40129, Italy *Correspondence to*: Matteo Rinaldi (m.rinaldi@isac.cnr.it)

**Abstract.** This study presents atmospheric ice nucleating particle (INP) data from the Gruvebadet (GVB) observatory in Ny-Ålesund (Svalbard). Aerosol particle sampling activities were conducted over three years (2018-2020), for a total of 6 intensive

- campaigns, covering three seasons (spring, summer and autumn). Ambient INP concentrations (*n*INP) were measured offline on the collected filters, in condensation freezing mode (water saturation ratio of 1.02), by means of the Dynamic Filter Processing Chamber (DFPC). Three activation temperatures (Ts) were considered: -15, -18 and -22°C. Overall, in the PM<sub>10</sub> size range, DFPC-measured *n*INP ranged from 0.3 to 315 m<sup>-3</sup> in the considered T range, in agreement with previous observations in the Arctic environment. Regarding the ice-nucleation efficiency of the investigated aerosol
- particles (referring to the range between 0.1 and 10  $\mu$ m), the estimated activated fraction (AF) resulted between 10<sup>-8</sup> and 10<sup>-5</sup>, obviously increasing as the T decreases.

The seasonality of the ice nucleating properties of Arctic aerosol particles was investigated by merging the results of the 6 campaigns. Our data show a moderate summertime increase of *n*INP at T = -15 °C. No such summertime increase was observed at T = -18 and -22 °C. On the other hand, the AF of atmospheric aerosol particles presents a clearer seasonal evolution, with

- maxima observed in late summer and early autumn. Finally, we report a marked seasonal evolution in the contribution of super-micrometer INPs. Coarse INPs increase significantly their contribution from spring (15-20%) to summer (~60%), while lower levels typically characterize the autumn season (20-50%). Our calculations also show that coarse particles have at least two orders of magnitude higher AF compared to sub-micrometre ones.
- The correlation with anthropogenic long range transport tracer black carbon, the contribution of ground types inferred from satellite data, the low-traveling back trajectory analysis and the aforementioned considerations regarding the varying seasonal contributions of sub- and super-micrometre INPs all indicate that the primary sources of springtime INPs at GVB are mostly located outside the Arctic. In contrast, local INP sources dominate during summer and early autumn. When land and sea are mostly free from snow and ice, both marine and terrestrial sources result important INP contributors at GVB. Regarding marine sources in particular, our analysis identifies potential marine INP sources located in the seawaters surrounding and immediately
- to the South of the Svalbard archipelago down to the waters around Iceland. Such sources apparently dominate *n*INP in summer and early autumn outside the major terrestrial INP bursts.

# **1** Introduction

The Arctic is one of the most climate-sensitive regions on Earth, undergoing warming at a rate 2-3 times the global average (Serreze and Barry, 2011; Wendisch et al., 2019) or even higher according to recent estimates (Rantanen et al., 2022). This

phenomenon, known as Arctic amplification, has relevant implications for global climate and depends on many factors. One of the main drivers is considered the positive surface albedo feedback (Screen and Simmonds, 2010; Hall, 2004) resulting from the reduction of Arctic sea ice extent (Stroeve et al., 2012; Serreze et al., 2007). Other drivers include atmospheric and oceanic heat transport from the mid-latitudes (Spielhagen et al., 2011), the greenhouse effect of additional water vapor (Graversen and Wang, 2009) and cloud feedbacks (Korolev et al., 2017; Vavrus, 2004; Intrieri et al., 2002). Cloud feedbacks

are particularly important for the Arctic climate given the ubiquity of Arctic stratiform clouds and their potential to affect the radiative balance at both the surface and the top of the atmosphere. Among the factors influencing Arctic clouds, ice-nucleating particles (INPs) play a critical role by initiating ice crystal formation, a process that governs cloud phase, optical properties, and lifetime (Murray et al., 2021).

INPs determine the microphysical properties of mixed-phase clouds, which, as said, dominate the Arctic atmosphere year-

- round. These clouds, containing both supercooled liquid droplets and ice crystals, influence surface energy budgets by altering radiation fluxes (Korolev et al., 2017; Morrison et al., 2012). For instance, the presence of ice crystals can increase cloud reflectivity, thereby cooling the surface, or promote precipitation, reducing cloud coverage and enhancing surface warming (Murray et al., 2021; Lohmann and Feichter, 2005). Recently, Carlsen and David (2022) documented the importance of INPs in mix-phase cloud formation, showing through satellite data that the availability of INPs is essential in controlling cloud phase
- evolution and that local sources of INPs in the high-latitudes play a key role in the formation of such clouds. The complex interplay between INP concentrations, cloud microphysics, and atmospheric dynamics makes their accurate representation in climate models essential for understanding Arctic feedback mechanisms (Storelvmo et al., 2011). Despite their importance, the sources, concentrations, and types of INPs in the Arctic remain poorly characterized, posing significant challenges to accurate climate predictions and contributing significantly to large uncertainties in climate models (Schmale et al., 2021; 55 Murray et al., 2021).

INPs are broadly classified into abiotic and biotic. Within the former category, mineral particles typically dominate below - 20°C (Hoose and Möhler, 2012); K-feldspar and quartz constitute notable exceptions as they can facilitate ice nucleation at higher T (Atkinson et al., 2013; Harrison et al., 2019). In contrast, biotic INPs, including bacteria, fungal spores, and marine biogenic particles, are often more active at warmer sub-zero temperatures (Morris et al., 2014; Murray et al., 2012), even

- though the ice nucleation efficiency of biotic INPs is highly variable (Kanji et al., 2017). Seawater was also identified as a source of biogenic INPs (Knopf et al., 2011; Wang et al., 2015; Wilson et al., 2015; Mccluskey et al., 2017). In the Arctic, marine biogenic INPs, linked to phytoplankton activity and sea spray aerosols, are of particular interest due to their potential to dominate INP activity during the open-water season (Mccluskey et al., 2018c; Irish et al., 2017; Ickes et al., 2020; Hartmann et al., 2021; Creamean et al., 2019).
- Since the beginning of INP explorations in the Arctic, the ocean was proposed as a potentially important source of INPs (Bigg, 1996; Bigg and Leck, 2001). More recently, Creamean et al. (2019) showed how biologically derived INPs were transported from deep Bering Strait waters to become airborne over the Arctic Ocean. Hartmann et al. (2021) presented indications that INPs at warmer temperatures (T > -15°C) are marine and locally emitted, by shipborne measurements close to Svalbard and in the vicinity of the ice edge. Inoue et a. (2021), during an Arctic research cruise on the marginal ice zone in the Chukchi Sea,
- observed warm and thermo-labile INPs increasing by 1 order of magnitude under high wave and strong wind conditions in comparison with the earlier period. According to Creamean et al. (2022), warm INPs observed in summertime over the high Arctic were likely from biological productivity in open water from the marginal ice zone. Eventually, Porter et al. (2022) associated high concentrations of heat-labile INPs over the North Pole (88-90°N) with air masses originating in the ice-free ocean environment off the Russian coast, with pack ice, open leads, and the marginal ice zone apparently being weaker sources.
- Other studies hypothesized that marine sources of INPs may be potentially relevant over the Arctic during summer, without reaching a conclusive evidence (Sze et al., 2023; Santl-Temkiv et al., 2019). On the other hand, important terrestrial INP sources have been highlighted in the Arctic by other studies, such as mineral dust from the glacial outwash plains in Svalbard (Tobo et al., 2019) or from Iceland deserts (Sanchez-Marroquin et al., 2020). Regarding biogenic terrestrial INPs, vegetation (Conen et al., 2016), runoff from watersheds (Tobo et al., 2019) and thawing
- permafrost (Barry et al., 2023; Creamean et al., 2020), together with bacterial productivity (Santl-Temkiv et al., 2019), have been identified as potential sources in the Arctic. Finally, the long-range transport from mid- to low-latitudes can be a nonnegligible source of INPs (Schmale et al., 2021; Vergara-Temprado et al., 2017).

Despite the increase of INP observations over the Arctic region occurred in the last few years, we still lack quantitative insights concerning the abundance, the properties and the sources of INPs in this complex environment. The present study aims to

- contribute to INP observations in the Arctic environment, investigating INP concentrations (*n*INP), annual trends and potential sources at the sea-level site of Gruvebadet (GVB; Ny-Ålesund, Svalbard). The dataset of INP observations object of the present work encompasses that already discussed in Rinaldi et al. (2021) (33 samples) and comprises four additional measurement campaigns, for a total of 113 samples spread over 3 years and covering 3 seasons (spring, summer and autumn). The ice nucleation efficiency of Arctic aerosol particles is also investigated by discussing their activated fraction (AF; Schrod et al.,
- 2020).

## 2 Methods

# 2.1 Aerosol sampling for offline INP analyses

Aerosol particle sampling was performed at the GVB observatory, located in proximity to the village of Ny-Ålesund (78°55' N, 11°56' E) on Spitsbergen, Svalbard. The observatory is located about 1 km southwest of the village, at 40 m above sea

level. Given the prevailing southerly winds, such location guarantees minimal influence by local pollution sources (Udisti et al., 2016).

Aerosol sampling occurred during six intensive campaigns spanning over three years. In the present work we define the seasons following Creamean et al. (2022): March to May, spring, June to August, summer and September to November, autumn. No measurements were performed during the Arctic winter, mostly for technical and logistical reasons. Two campaigns were held

- in 2018, one in spring (from 17 April to 2 May) and one in summer (from 11 to 27 July). The results of these first two campaigns have been extensively discussed in Rinaldi et al. (2021). Three campaigns were carried out in 2019, one in spring (12-23 April), one in summer (5-20 July) and a longer one in autumn, spanning from 4 October to 24 November. This last campaign occurred contextually to a larger INP investigation effort (the NASCENT campaign) which was described in Pasquier et al. (2022) and Li et al. (2023). Eventually, one last campaign occurred in autumn 2020 (15-26 September). In total, 113 samples were collected and analysed, 28 during spring, 33 in summer and 52 during autumn.
- Throughout all measurement campaigns, aerosol particles were sampled using nitrocellulose membrane filters (Millipore HABG04700, nominal pore size  $0.45 \mu m$ ) mounted in two parallel inlet systems: one configured with a PM1 size selector, and the other for PM10 (cut point in accordance with EN 12341, TCR Tecora). Both sampling lines operated at 38.3 (±2.0) L min<sup>-1</sup>. The height of the sampling inlets was set about 5m above ground level.
- Two samples per day—one from each inlet system—were collected, with each sampling event lasting between 3 and 4 hours. This short collection period was chosen to prevent particle overload on the filters. Samples were stored at ambient temperature until analysis.

## 2.2 INP measurements by the Dynamic Filter Processing Chamber

- All the samples were analysed using the membrane filter technique presented in Santachiara et al. (2010) and Rinaldi et al. (2017) within ca. 6 months from sampling. INP measurements occurred in condensation-freezing mode at a supersaturation with respect to water (S<sub>w</sub>) of 1.02 at three temperatures (Ts): -15, -18 and -22°C. Ice nucleation was visually evaluated by counting the number of ice crystals growing on individual aerosol particles on the sampled filter illuminated by a visible light source. The uncertainty in the DFPC-based INP assessment was estimated following Belosi et al. (2018) and Rinaldi et al.
- (2021) and resulted to be around  $\pm 30$  %. The instrumental background was evaluated by analysing blank filters at the same conditions as the samples. All the measurements were corrected for the filter background and the contribution of the filter background variability was integrated in the overall evaluation of *n*INP measurement uncertainty.

#### 2.3 Complementary measurements and analyses

#### 2.3.1 Meteorology

Meteorological parameters (T; pressure; relative humidity; wind speed) were provided by the Amundsen-Nobile Climate Change Tower positioned less than 1 km N–E of GVB (Mazzola et al., 2016), while precipitation data (type and amount) were taken from the eKlima database, provided by the Norwegian Meteorological Institute (https://seklima.met.no/observations/, last access: 21 September 2022).

# 2.3.2 Black carbon measurements

Evaluations of Equivalent Black Carbon (BC) were obtained at Gruvebadet through continuous online measurements carried out by means of a Particulate Soot Absorption Photometer (PSAP) (Gilardoni et al., 2023).

## 2.3.3 Air mass Back-trajectories

For each of the 113 samples collected throughout the 6 campaigns, two air mass back trajectory (BT) tracks were calculated (one at start of the sampling time and the other at the end of it). The BTs from the National Oceanic and Atmospheric

Administration (NOAA) HYSPLIT model (HYSPLIT4 with GDAS data: https://ready.arl.noaa.gov/, last access: 21 September 2022) (Stein et al., 2015; Rolph et al., 2017) were simulated for an altitude of 100 m above ground level at the GVB station with hourly backward time steps up to 5 days (120 h).

# 2.3.4 Satellite ground-type maps

Ground condition maps were obtained from the National Snow & Ice Data Center (NISDC; https://nsidc.org/, last access: 28
May 2022) Interactive Multisensor Snow and Ice Mapping System (IMS) (Helfrich et al., 2007; National Ice-Center, 2008) at a 4 km spatial resolution. The ground types considered are "seawater", "sea ice", "land", and "snow". "Seawater" indicates that the air mass travelled over the open ocean, while "sea ice" indicates passage over ice-covered waters. The "land" category represents air masses passing over land without snow cover, whereas "snow" denotes passage over snow-covered land. For each BT endpoint, we identified the corresponding ground type, considering only BTs that travelled at altitudes within the

145 boundary layer height; such height was extracted from the ECMWF-ERA5 dataset (Hersbach et al., 2020). Combining the information obtained along the whole BTs allowed estimating the contribution of each ground type to each INP sample.

## 2.3.5 Satellite chlorophyll-a data and correlation analysis

Similarly to Rinaldi et al. (2021), satellite-retrieved chlorophyll-*a* fields were used to track the evolution of oceanic biological activity in the Arctic Ocean during the measurement periods. The best estimate "cloud free" (Level-4) daily sea surface

chlorophyll-a concentration (CHL) data were downloaded from the EU Copernicus Marine Environment Monitoring Service (CMEMS; http://marine.copernicus.eu/, last access: 30 May 2022). The data product is available globally at ~ 4 km spatial resolution. From this global dataset, CHL fields were extracted in the Arctic Ocean during the campaign periods to be merged with INP data.

## 2.3.6 Concentration-weighted trajectory model

The concentration-weighted trajectory (CWT) method was used to determine the most probable source regions contributing to INP samples at GVB. For each sample, two BT tracks, one at the start and one at the end of the sampling period, were analysed to represent the pathways of incoming air masses. A comprehensive explanation of the applied equation and calculation procedures is available in Rinaldi et al. (2021). The trajectories were traced back over a 5-day period, with data points recorded at 1-hour intervals along each track.

#### 160 3 Results and Discussion

## 3.1 INP concentration and activated fraction at GVB

Figure 1 shows the overall *n*INP range observed at the GBV station across the six campaigns described above (2018-2020), while the time series of *n*INP for each campaign are available in Fig. S1 and S2. In the PM<sub>10</sub> size range, *n*INP ranges from 3.5 to 315.1 m<sup>-3</sup> (median 76.2), from 4.0 to 289.0 m<sup>-3</sup> (median 35.5) and from 0.3 to 193.3 m<sup>-3</sup> (median 14.3) at T= -22, -18 and -

- 15°C, respectively. Compilations of ground level Arctic *n*INP data can be found in Rinaldi et al. (2021) and Li et. (2022); more recent data collections have been presented also by Yun et al. (2022) and Conen et al. (2023). According to these data compilations, the overall range of *n*INP in the Arctic, in the T range between -15 and -22°C, is roughly between 10<sup>-1</sup> and 10<sup>3</sup> m<sup>-3</sup>, encompassing the totality of our data. It should be noted, anyhow, that comparison with these past studies is only qualitative given the great variability of parameters that may influence the measurement of *n*INP (e.g., different instruments,
- locations, season, weather conditions, aerosol particle size distribution, ice nucleation mode). In terms of activated fraction (AF), that is *n*INP scaled over the total particle number concentration in the 0.1 – 10 µm size range, the observed variability at GVB at T = -22°C goes between  $9.9 \times 10^{-8}$  and  $2.3 \times 10^{-5}$  (median  $2.4 \times 10^{-6}$ ; Fig. 2). At T =  $-18^{\circ}$ C AF ranges between  $5.8 \times 10^{-8}$  and  $1.7 \times 10^{-5}$  (median  $1.3 \times 10^{-6}$ ), while at T =  $-15^{\circ}$ C it ranges between  $9.0 \times 10^{-9}$  and  $7.4 \times 10^{-6}$  (median  $4.5 \times 10^{-7}$ ). Time resolved AF values for each campaign can be found in Fig. S3.
- Recently, data of aerosol particles AF at  $T = -15^{\circ}$ C, by immersion freezing, have been published by Li et al. (2023) for GVB station, from sampling occurred in parallel to one of the campaigns object of the present study (autumn 2019). The reported AF-15°C levels range approximately between  $3 \times 10^{-7}$  and  $2 \times 10^{-4}$ , between one and two orders of magnitude higher than in the present study. This discrepancy is due to the different operative definitions of AF used in the two studies. Indeed, Li et al. (2023) normalized *n*INP on the total particle number concentration from 500 nm of diameter, resulting in lower total particle
- counting and consequently higher AF. Apart from this quantitative discrepancy, the agreement of the AF temporal patterns between the two studies, during October-November 2019, is fairly good (Fig. S4).

#### 3.2 Seasonal evolution of ice nucleating properties

Figure 3 shows the seasonal evolution of *n*INP (PM<sub>10</sub> size range) reconstructed considering the results of the 6 campaigns. As the autumn 2019 campaign lasted almost two months, in the analysis it was considered as two separate periods, representing one the month of October and the other one the month of November. The assumption that the seasonality of INP parameters can be investigated by merging the results of campaigns performed over different years might be questioned; nevertheless, we evidence a very good agreement of the data distributions for the same season over different years (e.g., spring and summer 2018 vs spring and summer 2019 or autumn 2019 vs autumn 2020) which supports the assumption that our six campaigns provide a representative picture of the seasonal evolution of aerosol INP properties at the study location during the period

2018-2020.

In terms of seasonal evolution of *n*INP, different features can be observed for the three considered activation temperatures. At T = -22 and  $-18^{\circ}C$  we do not observe statistically significant differences between the INP levels in spring and summer, nevertheless we report a slightly higher median *n*INP in spring both at  $T = -22^{\circ}C$  (by 33%) and at  $T = -18^{\circ}C$  (by 17%). Conversely, autumn is characterized by a significant reduction of INP levels (p

independently on their direction, seasonal variations in *n*INP are lower than the day-to-day variability observed within each campaign.

A strong seasonal variation of *n*INP, with maxima in summertime and concentration over one order of magnitude higher than in the other seasons, has been often reported for the Arctic environment (Santl-Temkiv et al., 2019; Wex et al., 2019; Tobo et

- al., 2019). Recently, Creamean et al. (2022) confirmed these findings by ship-borne measurements in the high Arctic. Similarly, Sze et al. (2023) reported a marked summertime increase of *n*INP from two years of continuous measurements at Villum station in northern Greenland. In both studies, the seasonality was driven by the summertime enhancement of warm INPs (active at T>-15°C), attributed to local biological sources of marine (Creamean et al., 2022) or combined marine and terrestrial (Sze et al., 2023) origin. Sze et al. (2023) commented that the background of cold INPs (active at < -15°C, likely of mineral</p>
- or soil origin) progressively reduces the seasonal variation of the *n*INP at colder temperatures. This is in line with the finding presented here as also our GVB data show a summertime increase, even though less evident than in other works, for warm INPs ( $T = -15^{\circ}C$ ) and not for colder ones (T = -18 and  $-22^{\circ}C$ ). An explanation for this could be that Ny-Alesund is characterized by a higher cold INP background than other locations, maybe because of the relatively low latitude, which is able to mask the *n*INP seasonal trend also at  $T = -18^{\circ}C$ , temperature at which it is instead evident over the high Arctic (Creamean et al., 2022),
- and to reduce the extent of the summertime increase also at T = -15 °C. This would explain also why a negligible seasonal trend was reported also by Schrod et al. (2020) from multi-year measurements performed at Zeppelin station, on the peak overlooking GVB station (475 m asl). In fact, they report INP measurements at T = -20 °C and lower, which probably hindered the appreciation of the summertime *n*INP increase. Nevertheless, this hypothesis does not reconcile the discrepancy between our results and those by Wex et al. (2019), which showed a summertime increase of INPs by one order of magnitude or more
- at the same GVB sampling location, even though based on a much smaller data set. This difference might be explained by the interannual variability of meteorological conditions and aerosol particle sources influencing the ambient abundance of INPs at the sampling location. However, further studies would be necessary to derive any conclusive interpretation about the interannual variability of INP seasonal trends in the Artic.
- The seasonality of aerosol particles AF, as shown by Fig. 4, is not in phase with that of *n*INP, being generally characterized by a constant increase from spring to autumn. Furthermore, the seasonal variation of AF is stronger than that of *n*INP: at  $T = -22^{\circ}$ C, the median AF passes from  $1.3 \times 10^{-6}$  in spring to  $6.6 \times 10^{-6}$  in autumn (5.2 fold increase), with the autumn data being significantly higher than both spring and summer levels (p

270

measurements performed between March and October 2015. They showed that the particle number distribution is dominated by accumulation mode anthropogenic secondary aerosols in spring, while particle nucleation and biogenic secondary aerosols are more important in summertime, starting from May. Coarse mode particles peak in spring and autumn and are mainly contributed by sea-spray and blowing snow, with only a minor contribution from mineral particles. These seasonal patterns may contribute to explain the observed seasonal variation of the estimated AF. In springtime, the low AF can be explained

with the lower ice-nucleating ability of anthropogenic long range transported fine aerosols from lower latitudes (Hartmann et

- al., 2019). Conversely, in summer, notwithstanding the higher *n*INP observed at  $T = -15^{\circ}C$ , we do not observe a maximum in AF as the likely higher abundance of non-ice-nucleating particles from local NPF and growth keeps the AF at lower levels than in autumn. Finally, the reduction of secondary particles observed in autumn, probably due to the lower radiation characterizing that period, determines the enhancement of the AF observed in autumn, even though the absolute *n*INP tends to decrease with respect to summer.
- The NPF frequency in the Arctic atmosphere has been associated with the decreasing of sea ice extent (Dall'osto et al., 2019; Dall'osto et al., 2017; Dall'osto et al., 2018), probably via increased phytoplankton productivity, a phenomenon often indicated as Atlantification of the Arctic ocean. It is, therefore, likely that NFP impact will grow in the future. Similarly, the predicted reduction of snow and sea-ice coverage is likely to increase the Arctic *n*INP local sources. Predicting future *n*INP and aerosol particle AF over the Arctic in such a rapidly changing scenario is challenging. However, it provides the motivation for further 260 investigations in the region.

The strongest seasonal trend in INP properties was observed in the contribution of coarse INPs to the PM<sub>10</sub> INP pool (Fig. 5). Coarse INPs increase significantly their contribution from spring to summer, while lower levels typically characterize the autumn season. At  $T = -22^{\circ}C$  (-18°C), the median coarse INP contribution in summer is 43% (56%), as compared to 20% (15%) and 28% (19%) in spring and autumn, respectively. At  $T = -15^{\circ}C$  the contribution of coarse INPs passes from 22% in

- spring to 58% in summer, while the autumn level is 54% with high intra-season variability. Indeed, the median coarse INP contribution at T = -15°C in September is 73%, quickly decreasing through October (49%) and November (31%). This same trend is evident also at the other activation temperatures, even though to a lesser extent. This leads to the conclusion that a high contribution of coarse INPs may be a constant feature of the whole summer season, extending until late summer and early autumn (August-September), while it decreases progressively in October and November, reaching values as low as those
  - The very clear summertime increase in coarse INP contribution observed in the present study is consistent with the findings by Mason et al. (2016) at the Alert Arctic station and Creamean et al. (2022) over the high Arctic. Although size segregated INP measurements in the Arctic region are still scarce, this consistency between studies located in different sectors and at different latitudes of the Arctic region suggests that the increase in the contribution of coarse INPs occurring in summertime,
- likely attributable to local sources of potentially both marine and terrestrial origin, may be a general feature of the Arctic environment. This has implication on INP dynamics that need to be accounted for to model correctly Arctic clouds and climate.

## 3.3 Summer to autumn transition: focus on the 2019 campaign

The longest continuous record of INP properties in the present dataset is represented by the autumn 2019 campaign, which extended from the beginning of October to almost the end of November for a total of 40 samples. The sampling was performed

- within a larger experimental effort, which is described in Pasquier et al. (2022). Beside this, the 2019 observation period is interesting as it follows the transition from late summer to autumn conditions, never investigated by the DFPC before. Fig. 6 reports the time pattern of *n*INP in the PM<sub>10</sub> size range. The data show a distinct variability of *n*INP ranging between 3.5 and 185 m<sup>-3</sup> at T = -22°C, 4.2 and 74 m<sup>-3</sup> at T = -18°C and 0.6 and 43 m<sup>-3</sup> at T = -15°C. Incidentally, we report a very good level of agreement of this *n*INP ranges with those observed by Li et al. (2023) in parallel during the same period at GVB station by
- immersion freezing (Fig. S6).

characterizing the spring time.

290

Notwithstanding the day-to-day variability, *n*INP showed a clear decreasing trend through the campaign (statistically significant at  $T = -15^{\circ}C$ ; p

two sample subsets present statistically significant (p<0.05) differences in the *n*INP levels at all Ts (Fig. 8), with the landinfluenced subset having higher *n*INP both in terms of median and maximum values. This suggests a clear contribution of land sources in the study area during periods when snow is not present on the ground. The higher *n*INP associated with land-

- influenced air masses may be due to the higher ice nucleation efficiency of mineral dust and soil particles compared, for instance, to marine biological particles (Wilson et al., 2015; Mccluskey et al., 2018a; Mccluskey et al., 2018b). In summer, contacts with snow-free land occurred mainly over the Svalbard archipelago (local sources) or over Greenland and Iceland (regional sources), as shown in Fig. S3. Anyway, the low general influence of land sources in our sample set emerging from Figure 7 (i.e., only 15 samples over 113 show land influence above 5%) suggests that other sources may provide a significant
- contribution to the INP pool over the study area outside the major terrestrial dust outbursts.

# 3.4.3 Contribution of marine biological INP sources

In this Section, we test the hypothesis that marine sources of biological particulate matter contributed to the observed atmospheric INP pool in periods of our measurements when the sea was mostly free of ice. To do this, we operated a two-step approach, following Rinaldi et al. (2021). On the one hand, we examined the spatio-temporal correlation between *n*INP and surface CHL by applying the time-lag analysis firstly proposed by Rinaldi et al. (2013), to asses if INP levels followed any

surface CHL by applying the time-lag analysis firstly proposed by Rinaldi et al. (2013), to asses if INP levels followed any relation with the patterns of marine biological activity over a domain comprising the Arctic Ocean and all the seawaters down to 50° of latitude. On the other hand, we ran the CWT spatial source attribution model on the INP dataset in order to evidence INP emission hotspots on the same oceanic domain.

To exclude interferences from land sources, we used for the analysis the seawater-dominated sample subset defined in Sec.
3.4.2. In addition, we focused on INP measurements taken at T = -15°C, as this temperature is considered most indicative of ice nucleation driven by biological particles and less affected by mineral dust (Kanji et al., 2017). In this regard, Figure 8 clearly shows that the ratio between seawater-dominated and land-influenced samples is maximized (in terms of median values) at T = -15°C, which supports the above statement. Finally, we selected for the analysis the PM<sub>1</sub> size fraction following the results of McCluskey et al. (2018b) and Mansour et al. (2020b), where a better correlation with CHL is reported for fine 350 INPs.

An example of the results of the INP vs CHL correlation analysis is reported in Fig. 9a in the form of a correlation map. The colour of each pixel represents the correlation coefficient (R) resulting from the linear regression between the CHL concentration in that pixel and *n*INP measured at GVB. To explore potential correlations, we generated multiple maps by applying different time lags between the two time series. Specifically, CHL data were shifted from 1 to 24 days prior to the

- INP filter sampling times (see Figures S8, S9, and S10 for the complete set of lag-correlation maps). This time-lag methodology has been shown to enhance correlations between in situ coastal aerosol/cloud measurements and satellite-derived CHL fields (Rinaldi et al., 2013; Mansour et al., 2020a, 2020b, 2022). The lag period is thought to represent the timescale of biochemical processes that generate transferable organic matter in seawater following phytoplankton blooms indicated by CHL variability. Regions of the ocean displaying significant positive correlations—marked by red dots on the maps—may indicate source areas
- of biologically derived particles acting as INPs in our samples. The map in Fig. 9a shows high correlation regions in the seawaters to the South of the Svalbard archipelago, around Iceland and to the South of Greenland, all regions consistently located upwind of GVB during the sampling period (Fig. S3). Similar spatial features of the correlation between *n*INP and CHL can be observed almost independently on the considered lag times between 0 and 24 days (Fig. S8). This scarce dependence of the correlation on the lag time is typical of yearly (or multi-yearly) datasets and indicates that the relation
- between INP concentration and CHL is mainly dictated by the seasonal trend of marine biological activity, rather than by changes on shorter time scales (e.g., day-to-day or weekly).

The results of the CWT analysis are presented in Fig. 9b, which shows the potential INP sources at GVB during periods of high (sea-ice free) seawater influence. Fig. 9a and 9b show some similar features in terms of the identified potential source

regions. To facilitate the comparison between spatiotemporal correlation maps and the CWT results, Figure 9c shows every pixel with both a CWT value above the median and a significant positive correlation between *n*INP and surface CHL, considering every delay time between 0 and 24 days. The map in Figure 9c evidences the sea regions immediately to the south of Svalbard islands and around Iceland as the most likely sources of the INP measured at GVB in the seawater-influenced sample subset, i.e., outside the major episodes of terrestrial influence discussed in the previous Section.

The correlation with BC, the contribution of ground types, the BT analysis, and the aforementioned considerations regarding

- the varying seasonal contributions of sub- and super-micrometre INPs all indicate that the primary sources of springtime INPs at GVB are likely located outside the Arctic. Springtime INPs are thought to derive from lower-latitude regions together with anthropogenic aerosols, being transported northwards during the Arctic haze period (Stohl, 2006; Heidam et al., 1999). Conversely, the summertime aerosol particle population appears more related to local (Arctic) sources. Said sources tend to progressively reduce their contribution at the end of summer through autumn as directly observed during October and
- November 2019. Consequently, the AF estimates presented above support the hypothesis that long-range transported aerosol particles from lower latitudes nucleate ice less efficiently than local-origin aerosol particles, being spring the season characterized by the lowest AF. This aligns with the results reported by Hartmann et al. (2019), evidencing a minimal influence of human-induced emissions on Arctic INP levels, as evidenced by a comparison of present-day and pre-industrial values from ice core analyses, and with the pioneering study by Borys (1983).
- Our analysis points out that both marine and terrestrial sources may contribute to the INP population in the European Arctic. Terrestrial sources could play a significant role due to the greater ice-nucleating efficiency of mineral dust and soil particles compared to marine aerosols (Mccluskey et al., 2018b). By contrast, marine sources may be significant on account of the extension of ice-free seawaters during the Arctic summer and even in a future perspective, considering the progressive reduction of the sea-ice cover during the Arctic summer (Stroeve et al., 2012; Serreze et al., 2007). Regarding marine sources
- the analysis performed on the extended dataset (2018-2020) substantially confirms the preliminary results achieved by Rinaldi et al. (2021), showing potential marine sources located in the seawaters surrounding and immediately to the South of the Svalbard archipelago down to the waters around Iceland. These sea regions are identified by the statistical model CWT as INP emission hotspots and have time patterns of CHL evolution in correlating to some extent with the atmospheric variability of INPs at GVB. With respect to Rinaldi et al. (2021), the new results are based on a significantly higher number of samples (45)
- sample, more than 3 times higher) which provides more statistical robustness and credibility to the conclusions. Recently, Paglione et al. (2025) identified the main sources of the sub-micrometre organic aerosol at Ny-Ålesund by factor analysis of aerosol mass spectra and nuclear magnetic resonance (NMR) spectra. The study is based on aerosol samples from GVB station, collected between May 2019 and June 2020, therefore with a partial overlap with the present study (56 samples over 113). Among the isolated organic aerosol types, they identified a primary marine organic aerosol (POA), representative
- of biogenic organics emitted within sea-spray particles, which can be associated to marine biogenic INPs (Mccluskey et al., 2018a; Mccluskey et al., 2017). This organic aerosol type contributed mostly during summer. The sampling resolution of the filters analysed by Paglione et al. (2025) is of ca. 4 days, much longer than the that of the INP filters, which hinders a quantitative comparison with the present INP data. Nevertheless, the source area of the POA identified by CWT in Paglione et al. (2025) presents remarking similarities with Figure 9c of the present work. These findings mutually support each other,
- further strengthening the reliability of the identified marine biogenic INP source region. Although our dataset and analysis do not allow a quantitative assessment of the relative magnitude of terrestrial vs. marine INP sources over the European Arctic during periods of snow and sea-ice melt, our findings give support to the idea that marine biogenic particles may be a relevant INP source in the Arctic, in line with recent publications (Hartmann et al., 2021; Creamean et al., 2022; Inoue et al., 2021; Porter et al., 2022).