# Peer review of "Seasonal trends of Ice Nucleating Particles at Ny-Ålesund: a study of condensation-freezing by the Dynamic Filter Processing Chamber"

_Aerosol Research, 2025_

## Author Comment (AC1)

We thank the reviewers for their time and effort in evaluating our manuscript. We appreciate their constructive comments and suggestions, which have helped us to improve the clarity and robustness of the work. We have done our best to meet their requests. Even if not explicitly requested, given the general tenor of the comments, we decided to reformulate the title as follows: "Ice Nucleating Particles at Ny-Ålesund: a study of condensation-freezing by the Dynamic Filter Processing Chamber".

In the present document, original comments from the reviewers are reported in black, while answers are in blue.

**REVIEWER #1**

This study by Rinaldi et al. presents ice nucleating particle (INP) concentrations measured in spring, summer, and autumn over three years (2018, 2019, and 2020) at the Gruvebadet observatory in Ny-Ålesund, Svalbard.

While this manuscript presents valuable data and addresses an important topic, it faces significant methodological limitations that affect the reliability of its conclusions. With very brief daily sampling periods (3-4 hours) and uneven seasonal coverage (no data from winter, June, August, or March), the data presented in this study may not adequately capture the seasonal trends of Arctic INPs. Too many statements regarding agreements and correlations were vague and not supported by scientific quantifications, and the statistical approach lacks rigorous specification and justification. The manuscript motivations, objectives, and conclusions could also be strengthened by better summarizing findings from previous INP studies conducted in Ny-Ålesund and highlighting what new information is provided with the current study. I recommend that this manuscript should be reconsidered for final publication after major revisions.

**1. Major comments**

I strongly encourage the authors to have the manuscript thoroughly revised for language to improve some unclear sentences and the overall flow of the manuscript. I have noted a few specific language issues in my detailed comments, but a more comprehensive revision would help ensure that the scientific content is communicated more clearly and effectively.

Statistical tests and correlations evaluation:

Throughout the manuscript, p-values are reported without specifying which statistical tests were applied, specifically at: Lines 194, 197, 227, 297, and 327, and in Figure 6. This omission is particularly consequential considering that datasets like INP concentrations typically exhibit non-normal distributions spanning several orders of magnitude, where inappropriate tests can lead to invalid statistical conclusions. Thus, in addition to specifying which statistical tests were used, I strongly recommend the authors include justifications for the tests selected based on an analysis of the distribution of the INP concentration data.

We have now presented our statistical approach to testing differences between datasets more in detail in the new Sect. 2.3.7.

In our elaborations, we have used both the t-test (assuming normal distributions of the data) and the non-parametric Wilkoxon-Mann-Whitney test (not requiring normally distributed data). For 85% of the considered cases the outcomes were the same, suggesting that the normal distribution assumption was not so far from reality in many cases. However, to be conservative we decided to consider only the outcomes of the non parametric tests. For homogeneity, we now report through the new version of the text only the indication of the minimum tested significance level (p

The text was modified accordingly.

Line 292 "snow depth on the ground tends to be negatively associated to nINP": Did the authors quantify this association?

Please see the answer to the comment above.

Line 300 "a clear relation": Did the authors quantify such relation?

As above, the relation was inferred by visual analysis of Fig. 6. As such, we have rephrased the text to meet the reviewer's suggestions.

**Introduction:**

The introduction provides a good overview of previous INP studies conducted in the Arctic and their conclusions regarding the main sources of INPs in this region. However, considering the number of studies that have been conducted at the same sampling site as the results presented here, I recommend the authors add a paragraph focusing on INP measurements from Svalbard, specifically from Ny-Ålesund, to provide the readers with a clear picture of the current knowledge from this location. Such paragraph should summarize findings from studies which were cited by the authors in the Methods and Results sections:

Wex et al. (2019)

Schrod et al. (2020)

Rinaldi et al. (2021)

Pasquier et al. (2022)

Li et al. (2022)

Li et al. (2023)

As well as references which are currently missing:

Pereira Freitas et al. (2023)

Pereira Freitas et al. (2024)

Tobo et al. (2024)

In addition to summarizing the findings of these previous studies regarding the sources and seasonal cycle of INPs in Ny-Ålesund, the authors might want to highlight what knowledge gap their own study will cover.

A choice was operated in writing the Introduction of this work: we focused on evidencing the role of INPs in the Arctic climate system, describing which are the most relevant aerosol categories operating as

INPs in the atmosphere and presenting the main sources of INPs evidenced so far in the Arctic environment. We agree with the Reviewer that a focus on previous measurements in Svalbard could be fitting in such Introduction; we did not include such a paragraph in the original version to avoid being repetitive, considering that previous studies were cited and compared with our results in the following Sections. To meet the Reviewer's requests, we added a short paragraph summarizing previous studies at Svalbard (including the suggested missing references), leaving the detailed comparison with previous studies (when necessary) in the following Sections. We also added a few lines making explicit which are the specific gaps of knowledge that our work contributes to fill.

**Section 2.1:**

The brief daily sampling periods (3-4 hours) warrant additional considerations regarding potential diurnal biases in the collected INP dataset. While I understand that short collection periods are necessary to prevent particle overload on the filters, this approach creates a significant gap in capturing the diurnal variability of INPs. The manuscript provides no information regarding the time of day when sampling occurred (e.g., morning, afternoon, or evening) nor if the sampling times varied or were consistent across campaigns. Therefore, I strongly suggest that the precise sampling time windows be specified (the authors could add a Table in the Supplement). In addition, the authors should investigate if the systematic time-of-day effects influence the observed seasonal patterns, or at least acknowledge these limitations and discuss how this might influence interpretation of seasonal trends.

Sampling times depended in part on the necessity of coordinating with other activities scheduled at GVB station, therefore the sampling start time was not perfectly constant throughout the dataset. The sampling times for each sample are now reported in Table S1.

Given the characteristics of the sampling site, conspicuous daily patterns in INP and particle number concentrations are not expected. Indeed, the site is not influenced by anthropogenic activities, so there are no peaks associated with traffic rush hours or night time domestic heating as it happens in more anthropized environments. Furthermore, the high latitude reduces the daily excursion in radiation and boundary layer height for the majority of the time (in July there is always light, while from the end of October it is constantly dark). To investigate if the varying sampling times may have influenced the observed concentrations (impacting our inferred seasonality), we analysed the daily trends of particle number concentration in the following size ranges: from 100 nm to 10  $\mu$ m (N100), from 500 nm to 10  $\mu$ m (N500) and from 1000 nm to 10  $\mu$ m (N1000). The median daily trends reported in the plots below (Figure A1) typically do not show strong daily patterns in the particle number concentration, which appear relatively flat or showing fluctuations that appear more random than related to systematic features. This likely indicates the absence of strong daily trends for INPs as well and means that the selection of the daily sampling window could have hardly biased the INP measurements in a systematic way.

However, we recognize that taking 3-4 hour-long snapshots of the INP concentration to represent the daily average concentration is an approach not entirely free from risks and an appropriate caveat was inserted in the new version of the manuscript as suggested by the Reviewer:

"Given the need to coordinate with other scheduled activities at GVB, sampling could not be performed at the same time during all campaigns. Specific information on the sampling intervals is provided in Table S1. The relatively short (3 h) and variable sampling windows may have introduced biases in the quantification of INP levels during the single campaigns and, consequently, in the estimation of their

seasonal variability. However, analyses of continuous particle number concentrations did not reveal pronounced diurnal patterns, suggesting that potential biases arising from the variable sampling times were likely minimal".

(f)

Figure A1. Daily median and interquartile range of the particle number concentration in the size ranges from 100 nm to 10  $\mu$ m (N100), from 500 nm to 10  $\mu$ m (N500) and from 1000 nm to 10  $\mu$ m (N1000), during the (a) spring 2018, (b) summer 2018, (c) spring 2019, (d) summer 2019, (e) autumn (2019) and (f) autumn 2020 campaign.

Other factors limiting the interpretation of the results include:

Notable imbalances in seasonal coverage (28 samples from spring, 33 samples from summer, and 52 samples from autumn) which raises concerns about the representativeness of the data, especially given the authors' own acknowledgment that "seasonal variations in nINP are lower than the day-to-day variability observed within each campaign" (Lines 201-202). Short campaign durations (typically 2 weeks, with only the autumn 2019 campaign extending to ~2 months) provide limited capacity to capture intra-seasonal variability, which is particularly crucial during transition periods between seasons in the highly dynamic Arctic environment. Although I understand these limitations might be beyond the authors' control, they should be explicitly acknowledged in the manuscript.

We do not believe that our dataset is characterized by a "notable imbalance in seasonal coverage". The only season which is not covered by our dataset is winter and this is due to the logistical issues related to operating in an extreme polar environment in winter time. Indeed, no other study published so far and conducted at sea level in Svalbard (Wex et al., 2019; Rinaldi et al., 2021; Paquier et al, 2022; Li et al., 2022: 2023) has ever presented winter time data. Regarding the other seasons, each one contributed to the dataset for at least 25% of the data points, which we believe is enough to provide a seasonal picture. At this regard, we point out that the study by Wex et al. (2019), which is often cited for INP seasonality in the Arctic, infers the seasonality at Svalbard with 13 samples collected from March to September, for a total coverage of 52 days. Therefore, at least as far as regards measurements taken at sea level at Svalbard, the present work constitutes a significant step forward in terms of data coverage.

The complete absence of winter measurements creates a significant gap in the seasonal cycle characterization. I suggest the authors explicitly qualify claims about "seasonal trends" to acknowledge the missing winter component.

Please refer to the above comment. Explicit considerations about the missing winter season were added in the text: "No sampling could be conducted during winter due to logistical constraints. Therefore, throughout the text, any reference to seasonal cycles should be understood as describing the evolution of INP properties from spring through autumn, with winter being excluded".

Campaigns' timing within seasons: a close examination of the campaign dates reveals systematic timing patterns that may affect the representativeness of the seasonal characterizations: spring campaigns occurred exclusively during mid-to-late April through early May (missing March) and

summer campaigns were limited to July (missing June and August). The resulting seasonal patterns presented in this study may therefore reflect specific sub-seasonal phenomena rather than capturing the full seasonal variability. As for my previous comments, I suggest the authors explicitly acknowledge the possible sub-seasonal timing biases in each nominal season and discuss how this might affect the representativeness of seasonal characterizations. In addition, I recommend the authors be more specific when discussing period sampled, for example by using "late-spring" rather than simply "spring" to avoid overgeneralizing findings.

In our previous work (Rinaldi et al., 2021), we presented campaign based DFPC daily measurements in parallel with immersion freezing measurements (by WT-CRAFT) collected continuously from April to end of July 2018. The seasonality of INP described by the two different sampling approaches (limited to spring vs summer) was overall similar, nevertheless, it is true that the longer time coverage of immersion freezing measurements captured a variability that could not be observed relying only on the campaign-based DFPC observations.

Considerations about this limitations were added: "An additional limitation of this study arises from the fact that measurements did not span entire seasons but were instead restricted to short-term campaigns. This constrains our ability to resolve intra-seasonal variability and may, in turn, affect the robustness of our seasonality estimates. Rinaldi et al. (2021) compared campaign-based DFPC daily measurements with continuously collected immersion freezing data from April through late July 2018. While both approaches yielded broadly consistent descriptions of INP seasonality, the extended temporal coverage of immersion freezing measurements captured variability that could not be resolved using only campaign-based DFPC observations. For this reason, we advise readers to note that, hereafter, the term spring refers primarily to samples collected in April, whereas summer denotes samples collected in July. In contrast, autumn samples encompass a broader temporal range, spanning from September to November."

**Section 2.2:**

While the authors report a measurement uncertainty of approximately ±30% for INP concentrations (Line 120), this uncertainty is not visualized in figures, and insufficiently incorporated into seasonal comparisons. This is particularly problematic when comparing modest seasonal differences that approach the magnitude of the measurement uncertainty itself. I highly recommend the authors add error bars in Figures 1, 2, 3, 4, and 5 and discuss how these uncertainties affect the robustness of the observed seasonal changes.

Uncertainties associated to single data points are clearly showed were we believe they are most appropriated, i.e., in the time series plots provided in the supporting material. With respect to the present comment, we respectfully disagree with the Reviewers' point of view, for two reasons:

- 1) It would make no sense to place the error bars in the plots suggested by the Reviewer because they would just overlap each other resulting undistinguishable.
- 2) It would not be correct to interpret the seasonal variability (which we report as season median values) in light of the random uncertainties associated to each individual sample: random uncertainties can be assumed to compensate each other when the median of a sufficiently large number of data points (as in our case) is calculated. Median seasonal values would be affected, in case, by systematic errors, which we have no reason to assume to be present in the dataset, but not by random errors. To provide a framework for interpreting the variability of the seasonal median

values, we show and report their associated interquartile range and 5-95th percentile range (now also in the new Figure 2), which is more appropriate.

Section 2.3:

Important method information is missing in this section:

Section 2.3.1: Where was the precipitation data measured in Ny-Ålesund?

This information was provided: "Meteorological parameters (T; pressure; relative humidity; wind speed) were provided by the Amundsen-Nobile Climate Change Tower positioned less than 1 km N–E of GVB (Mazzola et al., 2016), while precipitation data (type and amount, measured in the centre of Ny-Ålesund by the Norwegian Meteorological Institute) were taken from ...".

Section 2.3.2: Could the authors provide more information regarding the black carbon measurements? What inlet was used? What flowrate? Were they any corrections applied to the data?

We provided more general information on PSAP measurements, referring to Gilardoni et al. (2023) for an extensive treatment of the data corrections.

Section 2.3.5: Information regarding the correlation analysis is missing (what was calculated and how).

Included in the new Sect 2.3.7 Statistical data treatment.

Please add a section describing the total particle number concentration measurements used to calculate the activated fraction.

Done (new Sect. 2.3.2 Black carbon and particle size distribution measurements).

Please add a section regarding the time-lag analysis conducted on surface CHL data.

The following details were added: "Recently, Mansour et al. (2020b) reported that *n*INP concentrations over the North Atlantic Ocean follow patterns of marine biological activity. To investigate the relationship between INPs and phytoplankton biomass (traced by CHL), daily DFPC samples were treated as individual days and compared with daily CHL time series, after selecting the samples more representative of marine sources (see Sect. 3.4.2). Pearson correlation coefficients between INPs and satellite-derived ocean colour data were calculated via least squares regression at each grid point in the Arctic domain across different time lags, generating the correlation maps shown in the Results".

Section 3.2:

The approach of merging data across multiple years (2018-2020) to characterize seasonality requires stronger statistical justification and assessment of inter-annual variability. While the authors claim "very good agreement of the data distributions for the same season over different years" (Line 187), they do not provide quantitative analysis to support this assertion. This aggregation approach is particularly concerning when the authors later invoke "interannual variability of meteorological conditions and aerosol particle sources" (Line 221) to explain discrepancies with Wex et al. (2019). In addition, the authors do not discuss how representative their dataset is relative to the seasonal cycle of INPs beyond their statement Lines 188-190 which is insufficient to quantify representativeness. I would recommend the authors present a more quantitative analysis of inter-annual variability to substantiate the claim of "very good agreement", and analyze meteorological and particle data over several seasons to determine how representative their own dataset is.

A quantitative analysis to support our statement of "very good agreement of the data distributions for the same season over different years" is now provided (Table S4). Briefly, we compared consistent periods over different years, namely April (spring) 2018 with April 2019 and July (summer) 2018 with July 2019 in terms of INP concentration. Median INP concentrations differ typically by less than 30% between the corresponding periods, with the exception of nINP-15°C in spring. In this case, the difference is by 73%. Statistical analysis of the data distribution was performed by the Wilkoxon-Mann-Whitney test, resulting in no statistically significant differences between the compared seasons. No such a quantitative comparison was possible for autumn as the two campaigns occurred in different periods (Oct-Nov in 2019, Sep in 2020). Nevertheless, we still believe that visual analysis of Figures 3 shows a clear consistency of INP levels between the 2019 and 2020 autumn campaigns. Also Figure 5 shows a progressive reduction of super-micrometre INP contribution from September to November which is consistent with the reducing trend evidenced across the longer 2019 campaign.

Regarding Line 221 of the old version, we thank the reviewer for pointing out this inconsistency in the manuscript, deriving from a previous version of the text: the text was modified removing any reference to interannual variability for which we do not have evidence.

According to the reviewer's suggestion we have extended the discussion about the seasonal representativity of our campaigns, providing a quantitative comparison between corresponding campaigns over different years (Table S4), nevertheless a detailed multi-year analysis of particle and meteorology data cannot be provided in this work, as it would go beyond the scopes of the manuscript also implying the necessity of retrieving data which are not available without costs to the Authors.

**Section 3.3:**

This section is titled "Summer to autumn transition" but only includes data from 4 October to 24 November 2019, which, as defined by the authors in the Methods section, correspond to autumn. Thus, I suggest that the authors modify the title of section 3.3 and rephrase any reference to a seasonal transition, including:

Line 289: "depicts a transition between late summer conditions"

Line 432: "contribution from late summer through autumn"

In addition to my previous comment regarding vague correlation statements which require more quantitative scientific descriptions, I would recommend the authors add a short conclusion at the end

of section 3.3 summarizing the main findings and putting them in perspective with the previous studies conducted at the same location at the same time during the NASCENT project. In particular, it would be useful to highlight what new information the current study provides compared to the findings from Li et al. (2023). Finally, a wide range of aerosol measurements were conducted during the NASCENT project, including data on particle chemical composition and fluorescence. Did the authors investigate potential relations between these parameters and their measured INP concentrations?

Title and Section 3.3 were modified: "3.3 Evolution of INP properties during autumn: a focus on the 2019 campaign".

The purpose of this paragraph within the manuscript's framework is to demonstrate that, during autumn, a clear trend can be observed in the main INP properties. This is already apparent from Figures 3, 4, and 5, and the purpose of this section is to further support our conclusions by showing that this trend is consistently observed sample by sample throughout the 2019 campaign, which is the longest available and spans approximately two months. Furthermore, we provide an interpretation of the main causes of the described trend (increased precipitation, increased snow cover and reduced contribution of local marine sources as sea-ice coverage increases). This is now made clearer in the new paragraphs added to this Stection: "This analysis further reinforces the conclusions discussed in the previous sections, revealing a distinct temporal evolution of INP properties (nINP, AF, and the relative contribution of super-micrometre INPs) across October and November. Overall, our findings (Figs. 3 - 6) suggest that summer-like conditions may persist into early autumn, as evidenced by the September 2020 campaign, whereas from October onwards, the atmosphere transitions toward a state characterized by reduced INP concentrations, lower aerosol particle AF, and a diminished super-micrometre INP contribution. Below, we provide an interpretative framework to explain the main drivers of this evolution". Considerations derived from this Section were also included in the new Section 3.4.4 (Remarks on INP sources): "Additionally, Figure 7 and Section 3.3 show that air masses had the highest contact with seawater in summer and early autumn, decreasing steadily through the fall. This implies that summertime INP sources may continue to influence the early autumn period, with their impact gradually diminishing later in the season, as clearly showed by the ca. two-month long record of observations of autumn 2019".

This section, by contrast, does not aim to provide additional information beyond the comprehensive and detailed work published by Li et al. (2023). First, said study is entirely focused on the autumn 2019 period, whereas our analysis aims to take a broader perspective by drawing on multiple (albeit short-term) observational datasets. Second, there was a significant difference in the resources available to the two teams during the 2019 campaign that would make this hardly possible for us. Similarly, a detailed comparison with the work by Li et al. (2023) is beyond the scope of this section and for this reason we would rather avoid making further comparison statements in this Section apart the short ones introduced in the revised version: "Incidentally, we report a very good level of agreement of this nINP ranges with those observed by Li et al. (2023) in parallel during the same period at GVB station by immersion freezing (Fig. S5). For instance, the Pearson's correlation coefficient (R) between the two nINP time series at T =  $-15^{\circ}$ C is 0.69, with a mean absolute error (MAE) of 13.1 m-3 and 84% of the points lying within a factor of 5 from the 1:1 line".

Finally, ancillary data such as fluorescent particle concentration data were not directly available to us therefore were not included. Aerosol chemistry data were analysed in Rinaldi et al. (2018): we decided to leave aerosol chemistry out of this work given the low contribution to INP source attribution seen in the previous work.

Why was the time spent in contact with land limited to 5% when considering land-influenced air masses? I understand that land contribution was very low throughout the measurements (Fig. 7), but this definition of land-influence air masses is very limited (up to 95% of the time could be spent over a different ground type) and bound to bias the results. I would recommend increasing the percentage, even if it means decreasing the size of the land-influenced subset.

We thank the reviewer for pointing out some weak points of this Section. For major clarity we revised it starting from the data analysis approach. In the revised manuscript, we comment now the difference in INP levels between two new sub-sets: the Land-influenced samples and all the rest of the dataset. The definition of the Land-influenced samples was not changed and was: all the "samples obtained from air masses in contact with land for more than 5% of the time". For this reason, the sub-set is defined Land-influenced and not Land-dominated. We operated this choice to show that even a modest contribution from land sources is able to significantly enhance the INP loading of the sample (as emerges from the revised Fig. 8).

To add some details, the following Table summarizes the distribution of the Land contribution associated to the samples, considering the whole dataset (second column) and considering only samples with a Land contribution > 0 (third column).

|                 | All samples | All samples with Land contribution >0 |  |
|-----------------|-------------|---------------------------------------|--|
| min             | 0.0         | 0.4                                   |  |
| 5th percentile  | 0.0         | 0.5                                   |  |
| 25th percentile | 0.0         | 1.5                                   |  |
| median          | 0.0         | 3.4                                   |  |
| 75th percentile | 1.7         | 7.4                                   |  |
| 95th percentile | 9.1         | 26.1                                  |  |
| max             | 33.3        | 33.3                                  |  |

Clearly, the Land contribution was generally very low, given the nature of the investigated environment. The selected threshold (5%) is already above the 75th percentile of the whole data distribution and above the median if we consider only samples with a non-negative Land contribution. This does not leave much room for selecting higher thresholds, which would have excluded the vast majority of the samples. Furthermore, using higher thresholds (e.g., 7.5 or 10%) we got spikes associated to the distribution of the "Non-Land-influenced sample" subset, associated to samples with Land influence >5% and lower than the new threshold. This strongly suggests that even a small time passed over land is able to affect the INP level towards high values.

To conclude, we believe that the selection of a "low" threshold (5% Land contribution) makes our message stronger: it is enough that the sampled air mass has passed over land for more than 5% of the time to see the effect as an enhancement of the INP loading in that sample. This suggests that terrestrial sources may be particularly important for INP levels in the Arctic, even though maybe in an episodic way.

If the authors proceed with the current definition, then please consider carefully adjusting your conclusions (as it is done adequately Lines 333-335), in particular at:

Line 28: "both marine and terrestrial sources result important INP contributors at GVB".

We are afraid there might be a misunderstanding here. In L333-335, we point to marine sources as potential alternative sources to terrestrial ones. This does not exclude other sources such as blowing snow over land or sea-ice, even though our analysis does not evidence a major role for this source with respect to Land inputs, marine sources and long range transport during Arctic haze (not shown here or in the text, as we believe this result would need more investigation to be fully reliable). In any case, we do not believe sentences such as that in L28 needs to be modified, as our conclusions are that both marine and terrestrial sources contribute INP in the Arctic.

Lines 327-328 "land-influenced subset having higher nINP" should be more carefully stated. In addition, how do the authors reconcile this with the findings from section 3.3 where higher INP concentrations were suggested to relate to air masses in contact with seawater?

We do not believe that L327-328 need to be rephrased: the land-influenced subset is characterized by higher nINP levels.

There is also no contradiction between this part and Sect 3.3. Discussing the Autumn 2019 campaign (Sect. 3.3), we showed that higher INP concentrations were associated to air masses in contact with seawater with respect to air masses passing more over Sea-ice or Snow. No comparison was done with samples in contact with (snow-free) Land as only three samples over the whole campaign show some land contribution during autumn 2019 and all in the first part of the campaign.

Lines 328-329 "clear contribution of land sources" should be carefully rephrased.

Line 385 "Our analysis points out that both marine and terrestrial sources may contribute to the INP population".

Line 439 "terrestrial sources resulted important INP contributors at GVB" should be more carefully stated.

Please see the above considerations for these comments as well. The above sentences have been modified to meet the reviewer's request.

| 2. | Minor | comr | nents |
|----|-------|------|-------|
|----|-------|------|-------|

Abstract:

Line 10: It might be worth briefly mentioning the two inlets used since sub and super-micrometer INPs are mentioned later in the abstract.

Done.

Line 24: I suggest listing the ground type in parenthesis (seawater, sea ice, land, and snow) to clarify.

**Done.**

Lines 30-31 "Such sources apparently dominate nINP in summer and early autumn outside the major terrestrial INP bursts": to which part of the manuscript is this referring to? To my understanding, the results presented in Figures 9, S8, S9, and S10 include data from each season sampled and thus cannot provide conclusions specific to summer or autumn period.

These considerations are supported by Fig. 7, showing that the majority of air mass contacts with seawater occurred in summer and early autumn (September).

**Section 2.2:**

I recommend adding more details regarding the analysis procedure with the Dynamic Filter Processing Chamber, as it was done in Rinaldi et al. (2021), including information regarding the sample preparation before analysis.

The requested details were added.

Lines 111 and 116: The authors mention that the filter samples were stored at ambient temperature up to 6 months after sampling. This raises concerns regarding the preservation of heat-labile biological INPs, which is particularly relevant for Arctic summer samples, where marine biological contributions are hypothesized to be important. Did the authors investigate the impact of such storage protocol on the INP content of the samples? If so it would be a valuable section to add to this manuscript. If not, the authors should explicitly acknowledge the potential impact of storage conditions on INP samples and discuss how this might affect the interpretation of seasonal patterns.

Unfortunately, such tests were not done. Actually, there is still no consensus in literature about the effect of storage conditions on INPs. As such we would prefer not to open this line of discussion for which non conclusion can be provided, having already clearly indicated how we stored the samples.

Line 117: Why did the authors select a supersaturation with respect to water of 1.02?

Operating in condensation freezing mode at  $S_w$  = 1.02 is the standard approach with the DFPC, even though higher supersaturations could be achieved as, for instance, in Belosi et al. (2018). A supersaturation of 1.01-1.02 is typically associated with atmospheric clouds (Pruppacher and Klett, 1997; DeMott et al., 2011) and this is the motivation of this choice.

Belosi et al. (2018), Tellus B, 70, 1–10, https://doi.org/10.1080/16000889.2018.1454809

Pruppacher, H. R. and Klett, J. D. 1997. Microphysics of Clouds and Precipitation, Kluwer Academic Publishers, Dordrecht, 954pp.

DeMott, P. J., Möhler, O., Stetzer, O., Vali, G., Levin, Z. and et al. . 2011. Resurgence in ice nuclei measurement research . Bull. Amer. Meteor. Soc. 92 , 1623 – 1635 . DOI: https://doi.org/10.1175/2011BAMS3119.1 .

Line 117: The limited temperature range (-15, -18, and -22 °C) examined in this study constrains the authors ability to fully characterize Arctic INPs, especially biological INPs that could be active at warmer temperatures. Although I understand this limitation is due to instrument constrains, it is particularly significant given the manuscript's focus on marine biogenic sources of INPs which often exhibit ice nucleation activity at temperatures warmer than -15 °C. Thus, I strongly recommend the authors acknowledge the limitation of the temperature range selected and discuss (here and in the Results section) how this might affect the ability to detect and characterize the full spectrum of biological INPs, particularly those of marine origin.

We respectfully disagree with the idea that our study's ability to detect marine biogenic INPs is significantly constrained by the selected temperature range (-15, -18, and -22 °C). INPs that activate at warmer temperatures (e.g., -5 or -10 °C) would still be active - and thus detectable - at -15 °C, if present in sufficient concentrations. Therefore, we do not consider our measurement range to have imposed a limitation in the detection of marine biological INPs per se. If any limitations exist, they might relate to our ability to distinguish between biogenic and mineral INPs due to the relatively narrow temperature range investigated. However, this remains difficult to demonstrate, given the variability reported in the literature regarding the freezing onset of biological INPs.

We also do not believe that adding a specific caveat regarding the temperature range is necessary. First, the temperatures at which measurements were conducted are clearly stated throughout the manuscript, so readers are fully informed of the experimental conditions. Second, we prefer to avoid speculating about the results we might have obtained had we used a different technique or targeted a broader temperature range, especially since such speculation would not be based on supporting data.

Line 118: How was the INP concentration derived from the number of ice crystals visually detected?

The number of crystal was divided by the volume sampled through the filter obtaining the INP number concentration per cubic metre. This was specified in the revised version.

Lines 121-122: Could the authors provide quantitative information regarding the background levels and their variability throughout the campaigns? In addition, how were the measurements corrected for the background and its variability exactly?

The background levels are now reported in a dedicated Table in the Supplementary material (Table S2).

According to the campaign, the DFPC background at the three activation temperatures was obtained measuring 3 non-sampled filters. The correction of the measurements for the background occurred in two steps:

- 1) the number of crystals counted on each sampled filter after the analysis was subtracted of the average number of background crystals (i.e., average of 3 non-sampled filters)
- 2) the standard deviation associated to the mean number of background crystals was accounted for in the INP uncertainty calculation by error propagation.

For the majority of the samples, the background level was small or even negligible with respect to the number of counted crystals.

The above information are now added to the test.

**Section 3.1:**

Lines 178-181: Did the author calculate the activate fraction using the same definition as Li et al. (2023) to confirm their statement?

Calculation of AF500 (AF estimated starting from a lower cut-off of 500 nm) was performed for the autumn 2019 campaign. The obtained AF range is between  $3.1 \times 10^{-7}$  and  $1.1 \times 10^{-3}$  in line with the findings by Li et al. (2023). We have reported this in the manuscript and added a Table in the Supporting (Table S3).

**Section 3.2:**

This section is lengthy and could be better structured to improve the overall flow. For example, the authors could consider adding some subsections for the seasonal evolution of the 1) PM10nINP (Lines 183-223), 2) AF (Lines 224-260), and 3) coarse INPs (Lines 261-276).

Done as suggested.

Lines 220-222: Was the interannual variability of meteorological conditions and aerosol particle sources investigated?

No it was not. This would go beyond the purpose of our study and, unfortunately, would require data which are not fully available to us.

Lines 240-252: The authors hypothesize a relationship between NPF and AF but do not present direct measurements of NPF events or their contribution to the particle population during the periods studied, which could be different than those reported in Song et al. (2022). Although this link between NPF and AF is plausible, I believe that it requires stronger evidence to move beyond speculation. I recommend adding available aerosol size distributions from the periods studied, or at least acknowledging more explicitly the speculative nature of the NPF explanation where direct evidence is lacking.

We fully agree that the link between AF and new particle formation (NPF) discussed in the manuscript is speculative, and we intended it as such. We believe that the current wording in the text clearly reflects this, avoiding definitive claims and framing the discussion as a plausible interpretation rather than a demonstrated mechanism. We do not believe that introducing a discussion on NPF in this manuscript would be appropriate. In fact, to our knowledge, there are no scientific papers where these two atmospheric research topics are addressed together and in detail. However, to strengthen our interpretation, we have added an additional reference that supports the seasonal pattern of NPF frequency at the sampling site.

Lines 255-260: This paragraph seems out of place. The first sentence regarding NPF should be moved to the previous paragraph, or deleted. The second and third sentences regarding predicted changes in the Arctic and potential impact of INPs should be moved elsewhere, perhaps in the conclusions?

This part of the text was removed.

Line 261: The definition of coarse INPs (size range between 1 and 10  $\mu$ m) should be clearly stated somewhere in the manuscript (at the moment it is only mentioned in the caption of Figure 5). In addition, I suggest adding a short sentence to explain how the coarse INP contributions (%) shown in Figure 5 were calculated.

In the revised version, we decided to use "super-micrometre" instead of "coarse". A definition of super-micrometre INPs and how we calculated them is now reported in the revised text: "The strongest seasonal trend in INP properties was observed in the contribution of super-micrometre INPs (Fig. 5), calculated as the difference between nINP in the PM10 and PM1 size ranges normalized for nINP in the PM10 size range".

Section 3.3:

Lines 293-294: I recommend highlighting the three periods defined in each panel of Figure 6 using vertical shaded bands.

Done.

Line 295: In my opinion, "growing levels of accumulated precipitations" is slightly misleading, or at least it should be mentioned that the second half of the second period is characterized by the lowest precipitations recorded over the whole campaign. Similarly, Line 304 "slight increase if precipitation" is only valid for the very first half of the second period.

The sentences were modified as: "In general, the three periods are characterized by growing levels of accumulated precipitations and snow cover depth, even though with variation of conditions within the single periods, as opposed to decreasing levels of *n*INP at all Ts"

and

"As seen above, the slight increase of precipitations with respect to the previous period (particularly in the first half of the second period) may also have contributed to reduce nINP".

Lines 291-297: Have the authors considered potential relation between INP concentrations and relative humidity?

Yes, no relation emerged.

Line 303: I would suggest rephrasing "resulting in a reduction of nINP" to "corresponding to a reduction of nINP", unless the authors have more evidence to show that it is indeed the change in air masses that caused the decrease in INP concentrations.

Modified as suggested.

Line 306: Please consider replacing "probably" by "potentially" or similar, unless the authors have further evidence of this.

Done.

Section 3.4.1

Could this sub-section be renamed "Correlation analysis between black carbon and INP concentrations" to be more accurate?

Done.

Lines 312-314: The sentence starting with "Consistently" is misleading as, to my knowledge, Rinaldi et al. (2021) did not show any BC data in their study. Please consider rephrasing.

Consistently refers to the fact that both datasets associate spring-time INPs to anthropogenic tracers: sulfate in the case of Rinaldi et al. (2021) and BC in the present work. The sentence has been made clearer in the revised version.

Section 3.4.3

Lines 344-350: Do I understand correctly that, other than limiting the data to PM1 INP concentrations recorded at -15 °C in seawater-dominated samples, the entirety of the dataset was used (i.e., data from all seasons studied)? If so, what motivated the authors to do so, instead of keeping the datasets segregated by seasons (spring, summer, and autumn) which would have given more information regarding the seasonal patterns?

The interpretation is correct. We decided to select the samples more impacted by potential seawater sources by an objective approach (percentual contact of the sampled air mass with ice-free seawaters over a threshold), avoiding arbitrary decisions as that of selecting only summertime samples. Obviously, the majority of the selected samples still derive from summertime measurements as in other periods contributions from sea-ice and snow tended to be higher.

Lines 348-350: How do the authors reconcile the choice of using PM1 INP data with the contribution of coarse INPs results presented in section 3.2 and Figure 5?

This was discussed in Rinaldi et al. (2021): "In our interpretation, the lack of a correlation between surface CHL concentration and coarse INPs does exclude the potential of the ocean surface to be a source of super-micrometer INPs. Rather, it simply shows that CHL is not the appropriate proxy to track the emission of large biological INPs from the oceans. Indeed, while CHL has previously been observed to correlate with the enrichment of organic matter in sub-micrometer sea spray (Rinaldi et al., 2013; O'Dowd et al., 2015), no investigation has ever been attempted with super-micrometer particles. In a laboratory-controlled setting, McCluskey et al. (2017) showed the production of both sub- and supermicrometer INPs (active at -22°C) from controlled algal blooms, pointing out that different particle types and production mechanisms are involved.

Some considerations were added in the revised manuscript as well.

Lines 363-366: I think the end of this sentence is a bit misleading and should be rephrased. The low dependence of the correlation on time lag indicates that the relation between INPs and CHL is not dictated by short-term changes of day scale, which the authors interpret as a hint that such relation is potentially related to seasonal trends of marine biological activity.

Precisely so. The sentence was rephrased as: "The weak dependence of the correlation on lag time is typical of yearly (or multi-yearly) datasets. This suggests that the relationship between INP concentration and CHL is primarily driven by seasonal patterns in marine biological activity, rather than by short-term fluctuations such as daily or weekly changes".

Starting from line 374: This reads like a summary/discussion and feels out of place in section 3.4.3. Could these paragraphs be included or merged with the conclusions?

These sentences were added in this part during the preliminary check of the manuscript on request of the Editor, which asked to present a Result and Discussion Section instead of a separate Discussion.

Lines 374-376: In section 3.4.2, the authors stated that "in spring, sampled airmasses had the majority of contacts with sea ice or snow-covered land". How is this interpreted as evidence that "sources of springtime INPS at GVB are likely located outside the Arctic" as written Lines 375-376? In addition, what do the authors mean by BT analysis here? If this is referring to the results presented in section 3.4.3, I do not understand how these results support the hypothesis that the sources of springtime INPs are likely located outside of the Arctic, since section 3.4.3 considers the entire period of dataset (limited to PM1/-15 °C data and seawater-dominated samples) and thus cannot provide conclusions specific to springtime. This comment applies to lines 24-27 of the abstract as well. If the authors use "BT analysis" to refer to Figure S7, I suggest they spend more time describing Figure S7 and its findings in the main text.

This piece of the text was originally part of a Discussion Section, which was integrated into the Results Section on request of the Editor during the preliminary manuscript evaluation. We realize from this comment that this came at the expense of clarity. The purpose of this section is to summarize our findings as a whole and does not regard Sect. 3.4.3 only. In order to make the text clearer, we have made this part in a separate sub-Section (Sect. 3.4.4 Remarks on INP sources) and placed references to the various parts of the text cited in this section.

Evidence that spring time INPs are mostly influenced by long range transport from lower latitudes is provided in Sect. 3.4.1: nINP correlates with BC only during spring time. It is also common knowledge that the Arctic experiences systematic long range transport of pollution from lower latitudes during spring time (appropriate citations are provided in the text). Furthermore, in Sect. 3.2, we show that springtime is characterized by the lowest contribution of super-micrometre INPs, which is consistent with a INP population coming from far away sources, as super-micrometre particles tend to be efficiently deposited during long range transport.

Line 378-380: Where in this manuscript is it shown that summertime aerosol particles appears more related to local sources which progressively reduce their contribution towards autumn? If this statement is based on literature, please add the necessary reference. In addition, the authors might want to indicate how this statement relates to INPs.

- (1) *n*INP does not correlate with BC in summer and autumn, this excludes major anthropogenic sources (Sect. 3.4.1).
- (2) *n*INP is mostly contributed by coarse particles in summer (Sect. 3.2) and such contribution is still quite high in early autumn reducing progressively through autumn (Sect. 3.3): this points to local sources as super-micrometre particles generally do not travel for long distances.
- (3) In Sect. 3.4.2 and 3.4.3 we show that potential sources of INPs during the period of minimum snow and sea-ice cover are potentially terrestrial sources (rarer occurrence but apparently these sources has a higher magnitude in producing INPs) and marine sources. For the latter we were able to pinpoint the most likely source locations, which were sea regions immediately to the south of Svalbard islands and around Iceland, i.e., local.
- (4) The majority of air mass contacts with seawater occur in summer and early autumn (Fig. 7), reducing in October and November as we show in Sect. 3.3.

All this shows that summertime aerosol particles appears related to local sources (natural, both marine and terrestrial) which progressively reduce their contribution towards autumn.

The text object of this comment and of the previous one now reads as follows:

"The correlation with BC observed only in springtime (Sect. 3.4.1) and the aforementioned dominance of sub-micrometre INPs in this season (Sect. 3.2) support the hypothesis that the primary sources of springtime INPs at GVB may be located outside the Arctic. During spring, the Arctic region is influenced by the long-range transport of anthropogenic aerosols originating from lower latitudes (Stohl, 2006; Heidam et al., 1999) and, during this time, INPs likely originate from the same source regions, carried northward with the Arctic haze. Consequently, the AF estimates presented above (Figure 4) support the hypothesis that long-range transported aerosol particles from lower latitudes nucleate ice less efficiently than local-origin aerosol particles, spring being the season characterized by the lowest AF. This aligns with the results reported by Hartmann et al. (2019), evidencing a minimal influence of human-induced emissions on Arctic INP levels in pre-industrial ice core records, and with the pioneering study by Borys (1983).

In contrast, the summertime aerosol population appears to be more strongly influenced by local Arctic sources. The absence of correlation with black carbon (Section 3.4.1) and the prevalence of supermicrometre INPs (Section 3.2) suggest that the dominant summertime INP sources are natural and located at relatively low distance. Further support comes from Sections 3.4.2 and 3.4.3, which indicate that during periods of minimal snow and sea-ice cover, terrestrial and marine sources with a local Arctic origin contribute to INP levels. Additionally, Figure 7 and Section 3.3 show that air masses had the highest contact with seawater in summer and early autumn, decreasing steadily through the fall. This implies that summertime INP sources may continue to influence the early autumn period, with their impact gradually diminishing later in the season, as clearly showed by the ca. two-month long record of observations of autumn 2019".

**Conclusions**

Lines 428-429: Where in this manuscript is it shown that coarse particles have a significantly higher AF compared to sub-micrometer ones? This comment applies to Lines 22-23 of the abstract as well.

This is clearly shown in Table 1 (same numbering in the old and revised versions of the text). In the revised version this part was moved to Sect 3.1.

Lines 433: Please consider rephrasing "can be explained by" by "could be related to" or similar, since this was not scientifically quantified.

Modified as suggested.

Lines 437: Where in the manuscript is it shown that local aerosol particle sources dominate during summer and early autumn? This comment applies to Line 27 of the abstract as well ("local INP sources dominate during summer and early autumn").

We believe now we have made this point clear. Evidence was showed in Sect. 3.2, 3.3 and 3.4 and specifically summarized in the new Sect. 3.4.4.

Lines 438: Could the authors specify which long-range transported aerosol particles they suggest are contributing to the INP population?

This is now clarified in the revised text (Sect. 3.4.4).

Overall: Could the authors spend some time highlighting what new information this manuscript provides, especially considering the few previous studies that were conducted at the same location over the same time periods, including their own study Rinaldi et al. (2021)?

This issue was addressed in the Introduction.

Figure 1: The spread in the data is not clear from such plots. I strongly recommend representing the data as boxplots (same for Figure 2). In addition, the PM10 and PM1 data could be presented in the same panel (with different colors) instead of having panels a) and b). This could be done for Figure 2 as well, and then Figure 1 and Figure 2 could be combined as sub-panels.

We decided to separate the plots for major clarity. Indeed, lumping  $PM_{10}$  and  $PM_1$  data as a function of T in the same plot would result in an unreadable overlapping of points or bars. We do not believe this could be beneficial for the clarity of the data presentation.

We tried box-plots as well as the current graphical version. Both versions has pros and cons, eventually we decided to leave the plots as they are, but using empty symbols for the single data points. This provides a clearer view of the data distribution and contextually evidences outliers in a clearer way. Using empty symbols was also suggested by Reviewer #2.

Figure 3: I recommend adjusting the y-axis scales for each panel to focus on the concentration range covered by the data at each temperature. This also applies to Figure 4.

Figures 3, 4 and 5 have been revisited graphically. Nevertheless, we would prefer to leave the y-axis scales as they are for consistency between the plots.

Figure 6: "Dark period" and "Polar night" are highlighted but never mentioned in the text. Consider removing or adding a mention in the main text. In addition, please explain how the dark period is defined exactly. In addition:

Removed.

How can the contribution of coarse INPs be zero around 26 October 2019?

In this period we observed a  $INP_{PM10}$  concentration as high as that of  $INP_{PM1}$ , resulting in the above mentioned coarse INP concentration equal or very close to zero. We do not have an explanation for this and do not believe it is central for the manuscript.

In panel f, I recommend plotting the data so that the bar width corresponds to the sampling duration of each INP sample.

Unfortunately this is not technically feasible given our resources.

In the caption, what do the authors mean exactly by correlation coefficients with respect to time?

The caption was modified into: "On the right, Pearson correlation coefficients between the observed variable and time (expressed as a sequential number) are reported to test the statistical significance of the observed temporal trend;...".

Figure 8b: Is there a specific reason why the datapoints are connected across temperatures?

I would say this was done to guide the eye.

Figure 9: Please add titles and units for each color bar.

Colour scale in panels (a) and (c) represent unitless values: (a) correlation coefficient and (c) number of occurrence of the conditions described in the caption. For panel (b) we have now provided the appropriate unit. Since not all the color scales are easily explained in just a few words of title, we prefer to direct the reader to the figure caption for a detailed explanation.

Figure S7: Could the "Day of the Year" be replaced by the actual dates, especially in 2019, to better follow the last paragraph of section 3.3?

Done.

Figure S8, S9, and S10: The color bars (scales) are missing.

| The colour scale is the same as in Figure 9a. It is stated in the caption.                                                                                                                      |
|-------------------------------------------------------------------------------------------------------------------------------------------------------------------------------------------------|
| 3. Technical comments                                                                                                                                                                           |
| Line 24: "transport" should be "transported"?                                                                                                                                                   |
| We do not think so. We modified into: long-range-transport tracer.                                                                                                                              |
| Line 25: what do the authors mean by "low-travelling"? low-level?  Yes. Corrected in low-level.                                                                                                 |
| Test. Corrected in tow tevet.                                                                                                                                                                   |
| Line 36: "is" should be moved after "considered" ("One of the main drivers considered is the positive []").                                                                                     |
| Done.                                                                                                                                                                                           |
| Line 49: do the authors mean "mixed-phase cloud formation" instead of "mix-phase"?  Corrected.                                                                                                  |
| Line 58: abbreviation "T" should probably be replaced by "temperatures" here.  Corrected.                                                                                                       |
| Line 61 and throughout the text: The second "c" in McCluskey is capitalized.                                                                                                                    |
| Thanks for catching this: it comes from the bibliography managing software. Corrected.                                                                                                          |
| Line 68: consider replacing "by shipborne" with "using shipborne".                                                                                                                              |
| Done.                                                                                                                                                                                           |
| Lines 69-71: I recommend moving the part of the sentence "during an Arctic research cruise on the marginal ice zone in the Chukchi Sea" to the end of the sentence to improve flow and clarity. |
| Done.                                                                                                                                                                                           |

Line 86: What do the authors mean by "object"? This sentence was reformulated. Line 103 "contextually": do the authors mean "concurrently with"? Corrected. Line 107: "PM" should be defined. Done. Line 115: I would recommend rephrasing the sentence to introduce the instrument name and its abbreviation: "All the samples were analysed using the membrane filter technique Dynamic Filter Processing Chamber (DFPC) presented in Santachiara et al. (2010) [...]". Done. Line 130: "Gruvebadet" should be "GVB" to be consistent with the notations used throughout the manuscript. Done. Line 135: define abbreviation HYSPLIT. Done. Line 155: Could the authors add the references corresponding to the concentration-weighted trajectory method? Rinaldi et al. (2021) is cited in the following lines as the reference for the methodology. Lines 158-159: At which altitude were the trajectories simulated? Was it also 100 m above ground level? Yes, as indicated in the Section on BT analysis.

Line 168 and throughout: Please avoid expressions such as "anyhow" and "anyway" and consider

using other words such as "however" when suitable.

Done.

Lines 175-176: I recommend rephrasing this sentence to improve the flow. For example: "Recently, AF measured at T = -15 °C in the immersion freezing mode have been published by Li et al. (2023) who conducted measurements at GVB station in parallel to one of the campaigns of the present study [...]".

Done.

Line 177: I recommend rephrasing "The reported AF-15°C levels range approximately between [...]" to "The AF reported at -15 °C range between approximately [...]".

Done.

Line 179: "normalized nINP on the total particle number concentration" should be normalized "by" or "with"?

We corrected using "by"

Line 200 "indeed": do the authors mean "instead"?

Corrected.

Line 215: Please consider moving "also" before "explain".

Done.

Line 216: Please consider removing "also" to avoid repetition.

Done.

Line 223: "Artic" should be "Arctic".

Done.

Line 229: The reference to Figure S5 (Line 232) could be moved to the end of this first sentence.

Done.

Line 281: Please consider replacing "by" by "with" in "never investigated by the DFPC [...]".

This lines were removed from the revised version.

Line 292 "In detail": Do the authors mean "In particular" or "More specifically"?

Modified in "more specifically".

Line 296: Please consider replacing "Anyhow" by "However". Done. Line 302: Is Fig. S3 the wrong reference here? Corrected. Line 312: Please consider adding "for" before "the spring time Arctic haze [...]" Done. Line 323: I recommend rephrasing the beginning of the sentence: "In order to assess the contribution of land and marine sources to the INP population, two subsets were isolated from the nINP dataset [...]". Done. Line 327: Please specify Fig. 8a. The whole Figure 8 shows higher nINP in Land-influenced samples therefore we would live the reference as it is. Line 333: Is Fig. S3 the wrong reference here? Corrected. Line 333: Please consider replace "Anyway" but "Nevertheless" or similar. Done. Line 340: "asses" should be replaced by "assess". Done. Line 344: Please consider moving "for the analysis" to the end of the sentence. This sentence was modified.

This sentence was removed as no more fitting the new version of the Figure.

Line 346: Please specify Fig. 8b.

Line 362: : Is Fig. S3 the wrong reference here?

Corrected.

Line 381: Please consider replacing "being spring" with "spring being".

Done.

Table 1: Caption should be changed from "Tabel" to "Table".

Done.

---

## Author Comment (AC2)

We thank the reviewers for their time and effort in evaluating our manuscript. We appreciate their constructive comments and suggestions, which have helped us to improve the clarity and robustness of the work. We have done our best to meet their requests. Even if not explicitly requested, given the general tenor of the comments, we decided to reformulate the title as follows: "Ice Nucleating Particles at Ny-Ålesund: a study of condensation-freezing by the Dynamic Filter Processing Chamber".

In the present document, original comments from the reviewers are reported in black, while answers are in blue.

**REVIEWER #2**

The manuscript by Rinaldi et al. presents a new dataset of ice-nucleating particle (INP) concentrations from the Gruvebadet observatory in Ny-Ålesund (Svalbard), covering spring, summer, and autumn periods during the years 2018–2020. This dataset is valuable, as aerosol–cloud interactions are of particular importance in the Arctic and are considered a potential mechanism contributing to the region's accelerated warming.

However, in my opinion, the overall quality of the study is compromised by a lack of robustness in the discussion of sample representativeness, as well as limited rigor in the application of statistical analyses and comparisons, which rely heavily on qualitative descriptions. These constitute the main methodological weaknesses of the manuscript.

The discussion of the results should be strengthened by making the connections between the presented data and the derived conclusions more explicit and well-supported. The manuscript would also benefit from a language revision and clearer articulation of its goals and main take-home message. While the development of new INP concentration datasets for the Arctic is undoubtedly valuable, it would be helpful for the authors to clarify what specific new insights their study contributes to the existing body of knowledge.

I believe the manuscript has potential, but major revisions are necessary before it can be accepted.

**MAJOR COMMENTS**

In this section, I provide major comments divided into two parts:

- General Comments: These refer to overarching issues that affect the manuscript as a whole, including concerns about sampling representativeness, statistical treatment, and the clarity of data interpretation.
- 2. **Section-Specific Comments:** These address issues related to individual sections of the manuscript, including suggestions for clarification, methodological details, and improvements in presentation.

**General**

**Sampling and representativeness of the study period**

The first issue concerns the reliability of the collected filters and whether the derived data can be considered representative of the seasons in which they were collected. I am skeptical that a two-week sampling period within each season is sufficient to yield seasonally representative results. Do the

authors have a justification for this choice? It may be useful to test the representativeness of such subperiods using other, longer datasets, to evaluate how well short sampling windows reflect broader seasonal patterns.

In our previous work (Rinaldi et al., 2021), we presented campaign based DFPC daily measurements in parallel with immersion freezing measurements (by WT-CRAFT) collected continuously from April to end of July 2018 (at 4-day resolution). The seasonality of INP described by the two different sampling approaches (limited to spring vs summer) was overall similar, nevertheless, it is true that the longer time coverage of immersion freezing measurements captured a variability that could not be observed relying only on the campaign-based DFPC observations. Other longer datasets are not available at Ny-Alesund, at least not at sea-level. Continuous datasets exist for the Zeppelin station, as reported in the introduction, but comparing such diverse sampling locations may lead to misinterpretation of the results. In the end, we outline that as far as regards sea-level measurements at Ny-Alesund, although with all its limitations, this work constitutes a significant step forward in terms of data coverage.

Considerations about these limitations were added in the revised text: "An additional limitation of this study arises from the fact that measurements did not span entire seasons but were instead restricted to short-term campaigns. This constrains our ability to resolve intra-seasonal variability and may, in turn, affect the robustness of our seasonality estimates. Rinaldi et al. (2021) compared campaign-based DFPC daily measurements with continuously collected immersion freezing data from April through late July 2018. While both approaches yielded broadly consistent descriptions of INP seasonality, the extended temporal coverage of immersion freezing measurements captured variability that could not be resolved using only campaign-based DFPC observations. For this reason, we advise readers to note that, hereafter, the term spring refers primarily to samples collected in April, whereas summer denotes samples collected in July. In contrast, autumn samples encompass a broader temporal range, spanning from September to November."

A second concern related to sampling representativeness is the limited daily sampling duration (only 3 to 4 hours per day) without discussion of expected intraday variability or justification for how representative these short time windows are of daily conditions. Additional explanation or analysis would strengthen the credibility of the dataset.

As stated in the manuscript, the sampling duration was constrained by the necessity of avoiding overloading the filters which would spoil the analysis. Unfortunately, logistical and resource limitations did not allow us to sample more than one filter per day in order to get a full coverage of the 24 hours. We have discussed this limitation of our dataset in the revised version of the manuscript. However, we can provide support to our working hypothesis that the 3-4 h snapshot we took with our sampling can be considered broadly representative of the daily concentrations.

First, given the characteristics of the sampling time, conspicuous daily patterns in INP and particle number concentrations are not expected. Indeed, the site is not influenced by anthropogenic activities, so there are no peaks associated with traffic rush hours or night time domestic heating as it happens in more anthropized environments. Furthermore, the high latitude reduces the daily excursion in radiation and boundary layer height for the majority of the time (in July there is always light, while from the end of October it is constantly dark). To assess if our sampling strategy may have influenced the observed concentrations (impacting our inferred seasonality), we analysed the daily trends of particle number concentration in the following size ranges: from 100 nm to 10  $\mu$ m (N100), from 500 nm to 10  $\mu$ m (N500) and from 1000 nm to 10  $\mu$ m (N1000). The median daily trends reported in the plots below typically do not show strong daily patterns in the particle number concentration, which appear relatively flat or showing (and small) fluctuations that appear more random than related to systematic features. This

likely indicates the absence of strong daily trends for INPs as well and means that the selection of the daily sampling window could have hardly biased the INP measurements in a systematic way.

"Given the need to coordinate with other scheduled activities at GVB, sampling could not be performed at the same time during all campaigns. Specific information on the sampling intervals is provided in Table S1. The relatively short and variable sampling durations (3–4 h) may have introduced biases in the quantification of INP levels during the single campaigns and, consequently, in the estimation of their seasonal variability. However, analyses of continuous particle number concentrations did not reveal pronounced diurnal patterns, suggesting that potential biases arising from the variable sampling times were likely minimal".

(e)

Figure A1. Daily median and interquartile range of the particle number concentration in the size ranges from 100 nm to 10  $\mu$ m (N100), from 500 nm to 10  $\mu$ m (N500) and from 1000 nm to 10  $\mu$ m (N1000), during the (a) spring 2018, (b) summer 2018, (c) spring 2019, (d) summer 2019, (e) autumn (2019) and (f) autumn 2020 campaign.

**Statistical tools and analysis**

Multiple comparisons are presented in this study, using varied metrics for quantification, when provided, for example: difference in % (e.g. line 193), reduction factor (line 195), correlation (e.g. line 311), fold increase (line 226). There is no test specified establishing the statistical significance, even though in most cases, p-values are provided. The level of significance is, however, not uniform throughout the manuscript. A mix of p-values (<0.01, <0.05 and <0.1) is used depending on the specific analysis being done. Additionally, agreements and correlations are often described in a qualitative way: the use of "agreement... fairly good" (line 181), "good agreement" (line 187), "very good level of agreement" (283), or "no clear correlation" (line 311) are some examples.

Metrics for comparisons have been uniformed through the text.

We have now summarized our statistical approach to testing differences between datasets more in detail in the new Sect. 2.3.7.

In our elaborations, we have used both the t-test (assuming normal distributions of the data) and the non-parametric Wilkoxon-Mann-Whitney test (not requiring normally distributed data). For 85% of the considered cases the outcomes were the same, suggesting that the normal distribution assumption was not so far from reality in many cases. However, to be conservative we decided to consider only the outcomes of the non-parametric tests. For homogeneity, we now report through the new version of the text only the indication of the minimum tested significance level (p<0.05) even in cases that resulted significant for higher confidence levels.

We have also removed or modified statements that where too qualitative through the text.

**Clarity of data interpretation**

The overall tone of the results discussion is somewhat convoluted and overly lengthy. In several instances, the inclusion of summary tables presenting key values and statistics would help condense

the information and improve readability. The main text could then focus on highlighting the general trends and key observations, rather than listing detailed numerical results. As it stands, it is difficult for the reader to retain or identify the main messages of the analysis.

We acknowledge that at times the discussion may appear complex; however, this is often inherent to the nature of scientific writing. We believe that presenting detailed numerical results is essential to substantiate our arguments and provide transparency in the interpretation of the data. Unfortunately, the need to discuss results obtained at various activation temperatures further adds to the complexity, but this is an unavoidable aspect when dealing with INP analysis. Nonetheless, we appreciate the suggestion and have included a summary table to help condense the content and improve readability (Table 2).

**Section-specific**

**Introduction**

The authors should mention the coarse particle contribution to the INP concentration. The analysis of both PM10 and PM1 filters should be justified and contextualized from the beginning.

A clearer presentation of the data gap or the specific type of analysis targeted by the authors would help to sharpen the study's purpose and make it more transparent to the reader.

We have highlighted that one of the contributions of this work is to provide "size segregated" INP information which are extremely rare in the Arctic.

"Furthermore, this work provides size segregated information on INPs, presenting the ice-nucleating capacity of sub-micrometre particles compared to super-micrometre ones and the relative contribution of such size classes to the INP pool in different seasons. Size segregated INP information is extremely rare in the Arctic environment, being provided by only a few other papers to the best of our knowledge (Mason et al., 2016; Creamean et al., 2022)".

**Methods**

Please review and consider the following gaps in the Methods section:

- A brief explanation of how INP concentrations are derived from the Dynamic Filter Processing Chamber measurements should be included.
- Information on particle number concentration measurements is missing and should be provided.
- The correlation analysis and statistical tools used for comparisons are not described and should be clearly outlined.
- Section 2.3.4 should be renamed to explicitly reflect the inclusion of ground-type contributions. Additionally, the methodology for assessing ground-type contributions should be clearly separated from the description of the satellite-derived ground maps and their categorization.

All the above requests have been addressed. Regarding the last point, to avoid breaking the text in too many very short sub-sections, we modified the title of Section 2.3.4 in "2.3.4 Satellite ground-type maps and ground type contribution calculation".

**Results and Discussion**

The authors state that, after analyzing the seasonal distributions of their data, they observe a
 "good agreement" that justifies merging data from different years by season to study trends.
 However, this agreement is not supported by any statistical analysis, or at least none is
 presented in the manuscript. Furthermore, it should be noted that only in 2018 and 2019 is
 there a partial overlap in the sampling periods for the campaigns classified as spring and
 summer.

A quantitative analysis to support our statement of "very good agreement of the data distributions for the same season over different years" is now provided (Table S4). Briefly, we compared consistent periods over different years, namely April (spring) 2018 with April 2019 and July (summer) 2018 with July 2019 in terms of INP concentration. Median INP concentrations differ typically by less than 30% between the corresponding seasons, with the exception of nINP-15°C in spring. In these case, the difference is by 73%, which is still modest. Statistical analysis of the data distribution was performed by the Wilkoxon-Mann-Whitney test, resulting in no statistically significant differences between the compared seasons. No such a quantitative comparison was possible for autumn as the two campaigns occurred in different periods (Oct-Nov in 2019, Sep in 2020). Nevertheless, we still believe that visual analysis of Figures 3 shows a clear consistency of INP levels between the 2019 and 2020 autumn campaigns. Also Figure 5 shows a progressive reduction of super-micrometre INP contribution from September to November which is consistent with the reducing trend evidenced across the longer 2019 campaign.

 According to the authors, "seasonal variations in nINP are lower than the day-to-day variability observed within each campaign". How does the interquartile range influence the comparison between seasons? Is it taken into account in the values provided?.

Seasonal variations were described using the median values of the data distributions as a simple metric to quantify eventual difference between seasons. Interquartile and max-min ranges are however clearly shown in the Figures (e.g., Fig. 3, 4 and 5) and now also in the new Table 2.

**MINOR COMMENTS**

**Introduction**

• Line 86: The authors should mention that the new dataset corresponds to INP concentrations measured in the condensation freezing mode. Additionally, a brief comment on the relevance of this mode for cloud formation in the Arctic would strengthen the context.

The requested comment was added.

"All the data presented in this work have been obtained by measuring INPs in condensation-freezing mode. Condensation-freezing may play a role in Arctic cloud formation, depending on ambient temperature and relative humidity. However, its relevance remains uncertain, as observational evidence is limited by the difficulty in distinguishing this mechanism from immersion freezing - processes that may, in some cases, be physically indistinguishable (Wex et al., 2014; Hiranuma et al., 2015)".

**Methods**

**Section 2.1**

 Have blank filters been collected to evaluate background signal levels? How were they collected and stored?

Blank filters were filters not exposed to sampling which travelled and were stored together with the sampled ones. More details on blanks and blank levels are now added to the revised text.

• Line 111: The authors note that the samples were stored at ambient temperature until analysis. However, storage at ambient temperature can affect the ice-nucleating ability of the collected aerosol particles, particularly those of biological origin. Have the authors tested how their storage protocol may influence the results, for instance, by comparing it with coldstorage conditions?

Unfortunately, such tests were not done. Actually, there is still no consensus in literature about the effect of storage conditions on INPs. As such we would prefer not to open this line of discussion for which non conclusion can be provided, having already clearly indicated how we stored the samples.

**Section 2.2**

• **Line 115:** Please briefly introduce the operating principle of the Dynamic Filter Processing Chamber (DFPC).

**Details were added.**

• **Line 117:** Can the authors justify why the selected temperatures and supersaturations were chosen for the analysis?

T = -22 and -15°C are the lowest and highest temperatures at which the DFPC can operate providing reliable results. T = -18°C was chosen as a convenient intermediate T setup.

Regarding supersaturation, operating in condensation freezing mode at  $S_w$  = 1.02 is the standard approach with the DFPC, even though higher supersaturations could be achieved as, for instance, in Belosi et al. (2018). A supersaturation of 1.01-1.02 is typically associated with atmospheric clouds (Pruppacher and Klett, 1997; DeMott et al., 2011) and this is the motivation of this choice.

Belosi et al. (2018), Tellus B, 70, 1–10, https://doi.org/10.1080/16000889.2018.1454809

Pruppacher, H. R. and Klett, J. D. 1997. Microphysics of Clouds and Precipitation, Kluwer Academic Publishers, Dordrecht, 954pp.

DeMott , P. J. , Möhler , O. , Stetzer , O. , Vali , G. , Levin , Z. and et al. . 2011 . Resurgence in ice nuclei measurement research . Bull. Amer. Meteor. Soc. 92 , 1623 – 1635 . DOI: https://doi.org/10.1175/2011BAMS3119.1 .

• **Line 120:** The reported uncertainty in DFPC-based INP concentrations is 30%. How was this uncertainty accounted for in the subsequent analysis of seasonal trends, for example?

The measurement uncertainty was considered when comparing single *n*INP data points. Nevertheless, it would not be correct to interpret the seasonal variability (which we report as season median values) in light of the random uncertainties associated to each individual sample: random uncertainties can be assumed to compensate each other when the median of a sufficiently large number of data points (as in our case) is calculated. Median seasonal values would be affected, in case, by systematic errors, which we have no reason to assume to be present in the dataset, but not by

random errors. To provide a framework for interpreting the variability of the seasonal median values, we show and report their associated interquartile range and 5-95th percentile range, which is most appropriate.

• **Line 121:** How was the background correction applied? Please specify the method used and provide appropriate citations.

According to the campaign, the DFPC background at the three activation temperatures was obtained measuring 3 non-sampled filters. The correction of the measurements for the background occurred in two steps:

- 1) the number of crystals counted on each sampled filter after the analysis was subtracted of the average number of background crystals (i.e., average of 3 or 4 non-sampled filters)
- 2) the standard deviation associated to the mean number of background crystals was accounted for in the INP uncertainty calculation by error propagation.

For the majority of the samples, the background level was small or even negligible with respect to the number of counted crystals.

The above information are now added to the test.

**Section 2.3**

• Line 126: Please specify the location from which the rain data was obtained.

Details were addedd: "Meteorological parameters (T; pressure; relative humidity; wind speed) were provided by the Amundsen-Nobile Climate Change Tower positioned less than 1 km N–E of GVB (Mazzola et al., 2016), while precipitation data (type and amount, measured in the center of Ny-Alesund by the Norwegian Meteorological Institute) were taken from the eKlima database (https://seklima.met.no/observations/, last access: 21 September 2022)".

• **Line 130:** What is the temporal resolution of the black carbon data? How were they temporally aligned with the INP observations?

Hourly BC data were used. We simply averaged the BC data in order to match the INP sampling time.

**Results and discussion**

**Section 3.1**

• Line 171: Please specify how particle number concentration was measured.

Done. A new Section was added (2.3.2 Black carbon and particle size distribution measurements).

• Line 179: Could the authors calculate the activated fraction (AF), as done in Li et al. (2023), to quantitatively assess the agreement? Additionally, a quantification of the discrepancy due to the use of different lower size cutoffs would be beneficial.

Calculation of AF500 was performed for the autumn 2019 campaign. The obtained AF range is between  $3.1 \times 10^{-7}$  and  $1.1 \times 10^{-3}$  in line with the findings by Li et al. (2023). We have reported this in the manuscript and added a Table in the Supporting (Table S3).

Nevertheless, we decided against adding further investigations of the discrepancy in AF due to the use of different lower size cutoffs in the text as this will go beyond the purposes of this study.

**Section 3.2**

The results and discussion throughout this section should be restructured and better
organized to improve readability. Several different aspects are addressed (such as the
seasonality of nINP, the activated fraction, and the contributions of fine and coarse particle
fractions to nINP). I recommend reorganizing the section to clearly distinguish and structure
the discussion around each of these key themes, including subsections if necessary.

The Section was dived into Sub-sections.

• **Line 187:** Please support the statement "very good agreement" regarding the data distributions with appropriate statistical evidence.

Done. New text was added and Table S4.

• Line 192: Clarify that the values being compared are medians.

It is already clearly stated in the following line: "nevertheless we report a slightly higher  $\underline{\text{median }}$  nINP in spring both at T = -22°C (by 33%) and at T = -18°C (by 17%)." If the reviewer refers to the statistical significance tests, they where not run on the medians but on the whole data distributions.

• Lines 192–200: Statistical significance should be assessed and reported for all the comparisons presented. In addition, please standardize the way seasonal changes are quantified—e.g., use either percentage change or increase/decrease factors consistently—or justify the use of different metrics.

Statistical significance tests are now all presented in the same way and the approach is described in Sect. 2.3.7. All the seasonal differences are now treated in the same way.

• **Line 200:** Please indicate where in the figures or data the snow and sea-ice melt is shown. Reference the specific figure, panel, or dataset.

This is a general consideration, based on our knowledge of the site: in July the land surface is free of snow and sea-ice extent is at its minimum.

• Line 209: Can the authors clarify what is meant by "background of cold INPs"?

This statement refers to the conclusions of Sze et al. (2023). According to their study, the difference in INP concentrations between winter and summer decreases with decreasing activation temperature, due to a persistent background of mineral dust particles present in the Arctic year-round. As a result, the seasonal increase in INPs during summer is evident at warmer activation temperatures, but becomes progressively less pronounced at colder ones. To account for the fact that we observe little to no summertime enhancement already at –18 °C, we suggest that the cold INP background may be even more dominant at Ny-Ålesund compared to more northerly locations. Sze et al. (2023) states that "it is likely that the observed background INPs originate from long-range transport," given that "local Arctic sources are sparse in winter, due to the surface being covered in snow and ice." This may help explain why Ny-Ålesund is particularly influenced by such transport processes, as it is located closer to lower-latitude, snow-free regions that are likely sources of these INPs.

A few more explanatory considerations were added to the text: "A common finding across all these studies is that the difference in nINP between summer and the rest of the year diminishes with

decreasing activation temperature. According to Sze et al. (2023), this trend is attributed to a persistent background of mineral dust particles (active at T < -15°C) present in the Arctic throughout the year, which progressively obscures the summertime increase at colder temperatures".

• **Line 210:** How do the studies cited compare quantitatively with the summertime increase reported by the authors?

Sze et al. (2023) report and increase of *n*INP at  $T=-15^{\circ}$ C of ~4 times in summer (quasi-snow-free months) than in the rest of the year, while it is ~1.4 at  $T=-22^{\circ}$ C.

This was added to the text.

For Creamean et al. (2022) only a visual analysis of Figure 4 was possible, due to problems in reading the data files provided with the paper. The following description was added: "Recently, Creamean et al. (2022) confirmed these findings by ship-borne measurements in the high Arctic, evidencing a remarkable increase in June and July, with respect to the rest of the year, for INPs active at T < -15 $^{\circ}$ C and a less evident increase for higher Ts".

**Section 3.3**

 This section seems more appropriate as a case study that could be incorporated under Section 3.2.

We have now divided Sect. 3.2 into sub-sections. We prefer to leave Sect. 3.3 as a separate section even though with a new title.

• Line 283: The authors mention a "good level of agreement" between their nINP ranges and those measured in parallel by Li et al. (2023) in immersion mode, supported by Figure S6. Please include a quantitative assessment of this agreement. How well do the two datasets align when compared as time series? Additionally, how different did the authors expect the datasets to be, given that different freezing modes were investigated?

We have provided some details to make our statement quantitative: "For instance, the Pearson's correlation coefficient (R) between the two nINP time series at T = -15°C is 0.69, with a mean absolute error (MAE) of 13.1 m-3 and 84% of the points lying within a factor of 5 from the 1:1 line". Nevertheless, being an instrumental intercomparison not the purpose of this work, we would like to keep this to a minimum level.

• Line 293 and Figure 6: The three study periods should be more clearly identifiable in Figure 6. As currently presented, the lines marking the "dark period" and "polar night" distract from clearly distinguishing the discussed periods.

**Done.**

• **Line 302:** The reference to Figure S3 appears to be incorrect. Figure S3 does not depict air mass trajectories.

Corrected.

**Section 3.4**

• Line 314 and Table 2: I strongly recommend revising the code to report statistical significance using a more standard notation, such as \*\* (p < 0.01), \* (p < 0.05), and † (p < 0.1). Note that p-values < 0.1 are sometimes interpreted as indicating "marginal significance."

**Done.**

• **Line 323:** Can the authors clarify whether these two subsets were taken from the entire study period?

Yes, indeed they were. This information was added.

• Line 326: Are there cases where airmasses spend more than 50% of their trajectory time over seawater but are still classified as land-influenced (i.e., over land for more than 5% of the time)?

Yes, there are. Indeed, this is the most typical case and the fact that the land influenced sub-group presents higher concentrations than all the rest of the samples and also of the sea-water dominated ones confirms that snow-free land is a powerful source of INPs.

• Line 327: Please refer to Figure 8a.

We are referring to the whole Figure 8 here.

• Line 333: Again, Figure S3 does not show the geographical locations of potential sources.

**Corrected.**

• Line 343: How are marine aerosol emission mechanisms taken into account?

CWT is a purely statistical approach which does not consider aerosol emission mechanisms.

• Line 348: Please briefly comment on why the fine size fraction of INPs correlates better with chlorophyll-a levels.

This was discussed in Rinaldi et al. (2021): "In our interpretation, the lack of a correlation between surface CHL concentration and coarse INPs does exclude the potential of the ocean surface to be a source of super-micrometer INPs. Rather, it simply shows that CHL is not the appropriate proxy to track the emission of large biological INPs from the oceans. Indeed, while CHL has previously been observed to correlate with the enrichment of organic matter in sub-micrometer sea spray (Rinaldi et al., 2013; O'Dowd et al., 2015), no investigation has ever been attempted with super-micrometer particles. In a laboratory-controlled setting, McCluskey et al. (2017) showed the production of both sub- and super-micrometer INPs (active at -22°C) from controlled algal blooms, pointing out that different particle types and production mechanisms are involved.

Some considerations were added in the revised manuscript as well.

• Line 362: Incorrect reference to Figure S3.

**Corrected.**

• **Line 365:** Can the authors provide an appropriate reference to support their interpretation regarding the lack of dependence on lag time?

Citations were added: Mansour et al. (2020a, b), Mansour et al. (2022). All of the already in the reference list of the paper.

- **Line 378:** Please specify the evidence on which the authors base their statement that the "summertime aerosol particle population appears more related to local (Arctic) sources."
- *n*INP does not correlate with BC in summer and autumn, this excludes major anthropogenic sources (Sect. 3.4.1).

- *n*INP is mostly contributed by coarse particles in summer (Sect. 3.2) and such contribution is still quite high in early autumn reducing progressively through autumn (Sect. 3.3): this points to local sources as super-micrometre particles generally do not travel for long distances.
- In Sect. 3.4.2 and 3.4.3 we show that potential sources of INPs during the period of minimum snow and sea-ice cover are potentially terrestrial sources (rarer occurrence but apparently these sources has a higher magnitude in producing INPs) and marine sources. For the latter we were able to pinpoint the most likely source locations, which were sea regions immediately to the south of Svalbard islands and around Iceland, i.e., local.
- The majority of air mass contacts with seawater occur in summer and early autumn (Fig. 7), reducing in October and November as we show in Sect. 3.3.

All this shows that summertime aerosol particles appears related to local sources (natural, both marine and terrestrial) which progressively reduce their contribution towards autumn.

The text object of this comment and of the previous one now reads as follows:

"The correlation with BC observed only in springtime (Sect. 3.4.1) and the aforementioned dominance of sub-micrometre INPs in this season (Sect. 3.2) support the hypothesis that the primary sources of springtime INPs at GVB may be located outside the Arctic. In spring, the Arctic receives long-range transport of anthropogenic aerosols from lower latitudes (Stohl, 2006; Heidam et al., 1999), during this time, INPs likely originate from the same source regions and are carried northward with the Arctic haze. Consequently, the AF estimates presented above support the hypothesis that long-range transported aerosol particles from lower latitudes nucleate ice less efficiently than local-origin aerosol particles, being spring the season characterized by the lowest AF. This aligns with the results reported by Hartmann et al. (2019), evidencing a minimal influence of human-induced emissions on Arctic INP levels, as evidenced by a comparison of present-day and pre-industrial values from ice core analyses, and with the pioneering study by Borys (1983).

In contrast, the summertime aerosol population appears to be more strongly influenced by local Arctic sources. The absence of correlation with black carbon (Section 3.4.1) and the prevalence of supermicrometre INPs (Section 3.2) suggest that the dominant summertime INP sources are natural and located at relatively low distance. Further support comes from Sections 3.4.2 and 3.4.3, which indicate that during periods of minimal snow and sea-ice cover, terrestrial and marine sources with a local Arctic origin contribute to INP levels. Additionally, Figure 7 and Section 3.3 show that air masses had the highest contact with seawater in summer and early autumn, decreasing steadily through the fall. This implies that summertime INP sources may continue to influence the early autumn period, with their impact gradually diminishing later in the season".

• Line 380: Please include an explicit reference to the AF mentioned (e.g., figure or table).

We referenced the reader to Figure 4 for an overview of the seasonal evolution of AF.

• **Line 383:** Please clearly connect the results reported by Hartmann et al. (2019) to the corresponding reference.

We are not sure we have interpreted this comment correctly, however we modified the sentence as follows: "This aligns with the results reported by Hartmann et al. (2019), evidencing a minimal influence of human-induced emissions on Arctic INP levels in pre-industrial ice core records". We have checked and the reference is correctly reported in the reference list.

**Conclusions**

• **Line 421:** Please clarify what the authors mean by "higher cold INP background" and better connect this statement to the supporting evidence.

Please refer to the above comments for a detailed discussion about the cold INP background.

This sentence was modified into: "This could be explained assuming a higher background concentration of cold INPs of likely mineral origin at GVB with respect to other stations located at higher latitudes. This may be able to somewhat mask the impact of summertime sources at T = -18 and -22°C".

• **Line 424:** This conclusion is trivial since the activated fraction (AF) depends on both INP and non-INP aerosols.

**This sentence was removed.**

• **Line 438:** Please elaborate on the potential sources related to the suggested long-range transport mechanism and the associated aerosol types.

We respectfully believe that this is not the right place for further elaborations. The association between spring time INPs and the Arctic haze phenomenon has already been addressed in the previous sections.

**Figures and tables**

• In Figures 1 and 2, I recommend using semi-transparent symbols for the "all data" points to better illustrate the distribution of individual data points. Alternatively, color-coding could be used to represent the density of observations.

**Done as suggested.**

• In Figures S1 and S2, the symbols corresponding to **T = -15 °C** appear smaller than the others; please ensure consistent symbol sizing across all temperatures.

**Done.**

• I suggest several modifications to Figures 3, 4, and 5 to improve clarity. First, I recommend using traditional box-and-whisker plots (similar to Figure 8a) and adjusting the plot widths to reflect the time period of the measurements. I also encourage adjusting the spacing of the x-axis ticks to make it more uniform. Finally, consider merging Figures 3, 4, and 5 into a single multi-panel figure to facilitate easier comparison.

Figures 3, 4 and 5 were modified following the reviewer's suggestions but we decided to keep them separated as it would result in a very complex multi-Figure otherwise.

• Table 1 could include row headers indicating min-max values. "Tabel" should be Table.

We reported that information in caption to keep the Table easy and readable.

• I suggest a few changes to Figure 8a to improve clarity. Temperatures could be placed on the x-axis. The number of colors can be reduced to two: one representing land-influenced and the other seawater-dominated conditions. These modifications would simplify the legend. Additionally, please clarify the meaning of the "x" inside the boxes.

We appreciate this suggestion, the figure was modified accordingly.

**TECHNICAL COMMENTS**

A thorough review of technical aspects has not been conducted. However, attention should be paid to the citation list to ensure consistent formatting throughout.

**References**

Li, G., Wilbourn, E., Cheng, Z., Wieder, J., Fagerson, A., Henneberger, J., Motos, G., Traversi, R., Brooks, S., Mazzola, M., China, S., Nenes, A., Lohmann, U., Hiranuma, N., & Kanji, Z. (2023). Physicochemical characterization and source apportionment of Arctic ice-nucleating particles observed in Ny-Ålesund in autumn 2019. Atmospheric Chemistry and Physics, 23, 10489–10516. https://doi.org/10.5194/acp-23-10489-2023.

Hartmann, M., Blunier, T., Brugger, S. O., Schmale, J., Schwikowski, M., Vogel, A., Wex, H., & Stratmann, F. (2019). Variation of ice nucleating particles in the European Arctic over the last centuries. Geophysical Research Letters, 46, 4007–4016. https://doi.org/10.1029/2019GL082311

**Citation**: https://doi.org/10.5194/ar-2025-13-RC2